# Optimal Rates for Averaged Stochastic Gradient Descent under Neural Tangent Kernel Regime

**Atsushi Nitanda**[1,2,3,†]**, Taiji Suzuki**[1,2,⋆]

[1]*Graduate School of Information Science and Technology, The University of Tokyo*
[2]*Center for Advanced Intelligence Project, RIKEN*
[3]*PRESTO, Japan Science and Technology Agency*
Email: [†]nitanda@mist.i.u-tokyo.ac.jp, [⋆]taiji@mist.i.u-tokyo.ac.jp

## Abstract

We analyze the convergence of the averaged stochastic gradient descent for over-parameterized two-layer neural networks for regression problems. It was recently found that a neural tangent kernel (NTK) plays an important role in showing the global convergence of gradient-based methods under the NTK regime, where the learning dynamics for overparameterized neural networks can be almost characterized by that for the associated reproducing kernel Hilbert space (RKHS). However, there is still room for a convergence rate analysis in the NTK regime. In this study, we show that the averaged stochastic gradient descent can achieve the minimax optimal convergence rate, with the global convergence guarantee, by exploiting the complexities of the target function and the RKHS associated with the NTK. Moreover, we show that the target function specified by the NTK of a ReLU network can be learned at the optimal convergence rate through a smooth approximation of a ReLU network under certain conditions.

## 1 Introduction

Recent studies have revealed why a stochastic gradient descent for neural networks converges to a global minimum and why it generalizes well under the overparameterized setting in which the number of parameters is larger than the number of given training examples. One prominent approach is to map the learning dynamics for neural networks into function spaces and exploit the convexity of the loss functions with respect to the function. The *neural tangent kernel* (NTK) (Jacot et al., 2018) has provided such a connection between the learning process of a neural network and a kernel method in a reproducing kernel Hilbert space (RKHS) associated with an NTK.

The global convergence of the gradient descent was demonstrated in Du et al. (2019b); Allen-Zhu et al. (2019a); Du et al. (2019a); Allen-Zhu et al. (2019b) through the development of a theory of NTK with the overparameterization. In these theories, the positivity of the NTK on the given training examples plays a crucial role in exploiting the property of the NTK. Specifically, the positivity of the Gram-matrix of the NTK leads to a rapid decay of the training loss, and thus the learning dynamics can be localized around the initial point of a neural network with the overparameterization, resulting in the equivalence between two learning dynamics for neural networks and kernel methods with the NTK through a linear approximation of neural networks. Moreover, Arora et al. (2019a) provided a generalization bound of $O(T^{-1/2})$, where $T$ is the number of training examples, on a gradient descent under the positivity assumption of the NTK. These studies provided the first steps in understanding the role of the NTK.

However, the eigenvalues of the NTK converge to zero as the number of examples increases, as shown in Su & Yang (2019) (also see Figure 1), resulting in the degeneration of the NTK. This phenomenon indicates that the convergence rates in previous studies in terms of generalization are generally slower than $O(T^{-1/2})$ owing to the dependence on the minimum eigenvalue. Moreover, Bietti & Mairal (2019); Ronen et al. (2019); Cao et al. (2019) also supported this observation by providing a precise

estimation of the decay of the eigenvalues, and Ronen et al. (2019); Cao et al. (2019) proved the *spectral bias* (Rahaman et al., 2019) for a neural network, where lower frequencies are learned first using a gradient descent.

By contrast, several studies showed faster convergence rates of the (averaged) stochastic gradient descent in the RKHS in terms of the generalization (Cesa-Bianchi et al., 2004; Smale & Yao, 2006; Ying & Zhou, 2006; Neu & Rosasco, 2018; Lin et al., 2020). In particular, by extending the results in a finite-dimensional case (Bach & Moulines, 2013), Dieuleveut & Bach (2016); Dieuleveut et al. (2017) showed convergence rates of $O(T^{\frac{-2r\beta}{2r\beta+1}})$ depending on the complexity $r \in [1/2, 1]$ of the target functions and the decay rate $\beta > 1$ of the eigenvalues of the kernel (a.k.a. the complexity of the hypothesis space). In addition, extensions to the random feature settings (Rahimi & Recht, 2007; Rudi & Rosasco, 2017; Carratino et al., 2018), to the multi-pass variant (Pillaud-Vivien et al., 2018b), and to the tail-averaging and mini-batching variant (Mücke et al., 2019) have been developed.

**Motivation.** The convergence rate of $O(T^{\frac{-2r\beta}{2r\beta+1}})$ is always faster than $O(T^{-1/2})$ and is known as the minimax optimal rate (Caponnetto & De Vito, 2007; Blanchard & Mücke, 2018). Hence, a gap exists between the theories regarding NTK and kernel methods. In other words, there is still room for an investigation into a stochastic gradient descent due to a lack of specification of the complexities of the target function and the hypothesis space. That is, to obtain faster convergence rates, we should specify the eigenspaces of the NTK that mainly contain the target function (i.e., the complexity of the target function), and specify the decay rates of the eigenvalues of the NTK (i.e., the complexity of the hypothesis space), as studied in kernel methods (Caponnetto & De Vito, 2007; Steinwart et al., 2009; Dieuleveut & Bach, 2016). In summary, the fundamental question in this study is

*Can stochastic gradient descent for overparameterized neural networks achieve the optimal rate in terms of the generalization by exploiting the complexities of the target function and hypothesis space?*

In this study, we answer this question in the affirmative, thereby bridging the gap between the theories of overparameterized neural networks and kernel methods.

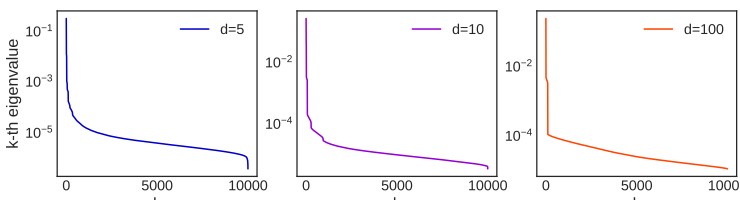

Figure 1: An estimation of the eigenvalues of $\Sigma_\infty$ using two-layer ReLU networks with a width of $M = 2 \times 10^4$. The number of uniformly randomly generated samples on the unit sphere is $n = 10^4$ and the dimensionality of the input space is $d \in \{5, 10, 100\}$.

## 1.1 CONTRIBUTIONS

The connection between neural networks and kernel methods is being understood via the NTK, but it is still unknown whether the optimal convergence rate faster than $O(T^{-1/2})$ is achievable by a certain algorithm for neural networks. This is the first paper to overcome technical challenges of achieving the optimal convergence rate under the NTK regime. We obtain *the minimax optimal convergence rates* (Corollary 1), inherited from the learning dynamics in an RKHS, for an averaged stochastic gradient descent for neural networks. That is, we show that smooth target functions efficiently specified by the NTK are learned rapidly at faster convergence rates than $O(1/\sqrt{T})$. Moreover, we obtain an explicit optimal convergence rate of $O\left(T^{\frac{-2rd}{2rd+d-1}}\right)$ for a smooth approximation of the ReLU network (Corollary 2), where $d$ is the dimensionality of the data space and $r$ is the complexity of the target function specified by the NTK of the ReLU network.

## 1.2 TECHNICAL CHALLENGE

The key to showing a global convergence (Theorem 1) is making the connection between kernel methods and neural networks in some sense. Although this sort of analysis has been developed in several studies (Du et al., 2019b; Arora et al., 2019a; Weinan et al., 2019; Arora et al., 2019b;

Lee et al., 2019; 2020), we would like to emphasize that our results cannot be obtained by direct application of their results. A naive idea is to simply combine their results with the convergence analysis of the stochastic gradient descent for kernel methods, but it does not work. The main reason is that we need the $L_2$-bound weighted by a true data distribution on the gap between dynamics of stochastic gradient descent for neural networks and kernel methods if we try to derive a convergence rate of population risks for neural networks from that for kernel methods. However, such a bound is not provided in related studies. Indeed, to the best of our knowledge, all related studies make this kind of connection regarding the gap on training dataset or sample-wise high probability bound (Lee et al., 2019; Arora et al., 2019b). That is, a statement "for every input data $x$ with high probability $|g_{\mathrm{nn}}^{(t)}(x) - g_{\mathrm{ntk}}^{(t)}(x)| < \epsilon$" cannot yield a desired statement "$\|g_{\mathrm{nn}}^{(t)} - g_{\mathrm{ntk}}^{(t)}\|_{L_2(\rho_X)} < \epsilon$" where $g_{\mathrm{nn}}^{(t)}$ and $g_{\mathrm{ntk}}^{(t)}$ are $t$-th iterate of gradient descent for a neural network and corresponding iterate described by NTK, and $\|\cdot\|_{L_2(\rho_X)}$ is the $L_2$-norm weighted by a marginal data distribution $\rho_X$ over the input space. Moreover, we note that we cannot utilize the positivity of the Gram-matrix of NTK which plays a crucial role in related studies because we consider the population risk with respect to $\|\cdot\|_{L_2(\rho_X)}$ rather than the empirical risk.

To overcome these difficulties we develop a different strategy of the proof. First, we make a bound on the gap between two dynamics of the averaged stochastic gradient descent for a two-layer neural network and its NTK with width $M$ (Proposition A), and obtain a generalization bound for this intermediate NTK (Theorem A in Appendix). Second, we remove the dependence on the width of $M$ from the intermediate bound. These steps are not obvious because we need a detailed investigation to handle the misspecification of the target function by an intermediate NTK. Based on detailed analyses, we obtain a faster and precise bound than those in previous results (Arora et al., 2019a).

The following is an informal version of Proposition A providing *a new connection* between a two-layer neural networks and corresponding NTK with width $M$.

**Proposition 1** (Informal). *Under appropriate conditions we simultaneously run averaged stochastic gradient descent for a neural network with width of $M$ and for its NTK. Assume they share the same hyper-parameters and examples to compute stochastic gradients. Then, for arbitrary number of iterations $T \in \mathbb{Z}_+$ and $\epsilon > 0$, there exists $M \in \mathbb{Z}_+$ depending only on $T$ and $\epsilon$ such that $\forall t \leq T$,*

$$\|\overline{g}_{\mathrm{nn}}^{(t)} - \overline{g}_{\mathrm{ntk}}^{(t)}\|_{L_\infty(\rho_X)} \leq \epsilon,$$

*where $\overline{g}_{\mathrm{nn}}^{(t)}$ and $\overline{g}_{\mathrm{ntk}}^{(t)}$ are iterates obtained by averaged stochastic gradient descent.*

This proposition is the key because it connects two learning dynamics for a neural network and its NTK through overparameterization without the positivity of the NTK. Instead of the positivity, this proposition says that overparameterization increases the time stayed in the NTK regime where the learning dynamics for neural networks can be characterized by the NTK. As a result, the averaged stochastic gradient descent for the overparameterized two-layer neural networks can fully inherit preferable properties from learning dynamics in the NTK as long as the network width is sufficiently large. See Appendix A for detail.

### 1.3 ADDITIONAL RELATED WORK

Besides the abovementioned studies, there are several works (Chizat & Bach, 2018b; Wu et al., 2019; Zou & Gu, 2019) that have shown the global convergence of (stochastic) gradient descent for overparameterized neural networks essentially relying on the positivity condition of NTK. Moreover, faster convergence rates of the second-order methods such as the natural gradient descent and Gauss-Newton method have been demonstrated (Zhang et al., 2019; Cai et al., 2019) in the similar setting, and the further improvement of Gauss-Newton method with respect to the cost per iteration has been conducted in Brand et al. (2020).

There have been several attempts to improve the overparameterization size in the NTK theory. For the regression problem, Song & Yang (2019) has succeeded in reducing the network width required in Du et al. (2019b) by utilizing matrix Chernoff bound. For the classification problem, the positivity condition can be relaxed to a separability condition using another reference model (Cao & Gu, 2019a;b; Nitanda et al., 2019; Ji & Telgarsky, 2019), resulting in mild overparameterization and generalization bounds of $O(T^{-1/2})$ or $O(T^{-1/4})$ on classification errors.

For an averaged stochastic gradient descent on classification problems in RKHSs, linear convergence rates of the expected classification errors have been demonstrated in Pillaud-Vivien et al. (2018a);

Nitanda & Suzuki (2019). Although our study focuses on regression problems, we describe how to combine their results with our theory in the Appendix.

The mean field regime (Nitanda & Suzuki, 2017; Mei et al., 2018; Chizat & Bach, 2018a) that is a different limit of neural networks from the NTK is also important for the global convergence analysis of the gradient descent. In the mean field regime, the learning dynamics follows the Wasserstein gradient flow which enables us to establish convergence analysis in the probability space.

Moreover, several studies (Allen-Zhu & Li, 2019; Bai & Lee, 2019; Ghorbani et al., 2019; Allen-Zhu & Li, 2020; Li et al., 2020; Suzuki, 2020) attempt to show the superiority of neural networks over kernel methods including the NTK. Although it is also very important to study the conditions beyond the NTK regime, they do not affect our contribution and vice versa. Indeed, which method is better depends on the assumption on the target function and data distribution, so it is important to investigate the optimal convergence rate and optimal method in each regime. As shown in our study, the averaged stochastic gradient descent for learning neural network achieves the optimal convergence rate if the target function is included in RKHS associated with the NTK with the small norm. It means there are no methods that outperform the averaged stochastic gradient descent under this setting.

## 2 PRELIMINARY

Let $\mathcal{X} \subset \mathbb{R}^d$ and $\mathcal{Y} \subset \mathbb{R}$ be the measurable feature and label spaces, respectively. We denote by $\rho$ a data distribution on $\mathcal{X} \times \mathcal{Y}$, by $\rho_X$ the marginal distribution on $X$, and by $\rho(\cdot|X)$ the conditional distribution on $Y$, where $(X, Y) \sim \rho$. Let $\ell(z, y)$ ($z \in \mathbb{R}, y \in \mathcal{Y}$) be the squared loss function $\frac{1}{2}(z - y)^2$, and let $g : \mathcal{X} \to \mathbb{R}$ be a hypothesis. The expected risk function is defined as follows:

$$\mathcal{L}(g) \overset{def}{=} \mathbb{E}_{(X,Y) \sim \rho}[\ell(g(X), Y)]. \tag{1}$$

The Bayes rule $g_\rho : \mathcal{X} \to \mathbb{R}$ is a global minimizer of $\mathcal{L}$ over all measurable functions.

For the least squares regression, the Bayes rule is known to be $g_\rho(X) = \mathbb{E}_Y[Y|X]$ and the excess risk of a hypothesis $g$ (which is the difference between the expected risk of $g$ and the expected risk of the Bayes rule $g_\rho$) is expressed as a squared $L_2(\rho_X)$-distance between $g$ and $g_\rho$ (for details, see Cucker & Smale (2002)) up to a constant:

$$\mathcal{L}(g) - \mathcal{L}(g_\rho) = \frac{1}{2}\|g - g_\rho\|^2_{L_2(\rho_X)},$$

where $\| \cdot \|_{L_2(\rho_X)}$ is $L_2$-norm weighted by $\rho_X$ defined as $\|g\|_{L_2(\rho_X)} \overset{def}{=} \left( \int g^2(X) \mathrm{d}\rho_X(X) \right)^{1/2}$ ($g \in L_2(\rho_X)$). Hence, the goal of the regression problem is to approximate $g_\rho$ in terms of the $L_2(\rho_X)$-distance in a given hypothesis class.

**Two-layer neural networks.** The hypothesis class considered in this study is the set of two-layer neural networks, which is formalized as follows. Let $M \in \mathbb{Z}_+$ be the network width (number of hidden nodes). Let $a = (a_1, \ldots, a_M)^\top \in \mathbb{R}^M$ ($a_r \in \mathbb{R}$) be the parameters of the output layer, $B = (b_1, \ldots, b_M) \in \mathbb{R}^{d \times M}$ ($b_r \in \mathbb{R}^d$) be the parameters of the input layer, and $c = (c_1, \ldots, c_M)^\top \in \mathbb{R}^M$ ($c_r \in \mathbb{R}$) be the bias parameters. We denote by $\Theta$ the collection of all parameters $(a, B, c)$, and consider two-layer neural networks:

$$g_\Theta(x) = \frac{1}{\sqrt{M}} \sum_{r=1}^M a_r \sigma(b_r^\top x + \gamma c_r), \tag{2}$$

where $\sigma : \mathbb{R} \to \mathbb{R}$ is an activation function and $\gamma > 0$ is a scale of the bias terms.

**Symmetric initialization.** We adopt symmetric initialization for the parameters $\Theta$. Let $a^{(0)} = (a_1^{(0)}, \ldots, a_M^{(0)})^\top$, $B^{(0)} = (b_1^{(0)}, \ldots, b_M^{(0)})$, and $c^{(0)} = (c_1^{(0)}, \ldots, c_M^{(0)})^\top$ denote the initial values for $a$, $B$, and $c$, respectively. Assume that the number of hidden units $M \in \mathbb{Z}_+$ is even. The parameters for the output layer are initialized as $a_r^{(0)} = R$ for $r \in \{1, \ldots, \frac{M}{2}\}$ and $a_r^{(0)} = -R$ for $r \in \{\frac{M}{2} + 1, \ldots, M\}$, where $R > 0$ is a positive constant. Let $\mu_0$ be a uniform distribution on the sphere $\mathbb{S}^{d-1} = \{b \in \mathbb{R}^d \mid \|b\|_2 = 1\} \subset \mathbb{R}^d$ used to initialize the parameters for the input layer. The parameters for the input layer are initialized as $b_r^{(0)} = b_{r+\frac{M}{2}}^{(0)}$ for $r \in \{1, \ldots, \frac{M}{2}\}$, where $(b_r^{(0)})_{r=1}^{\frac{M}{2}}$

are independently drawn from the distribution $\mu_0$. The bias parameters are initialized as $c_r^{(0)} = 0$ for $r \in \{1, \ldots, M\}$. The aim of the symmetric initialization is to make an initial function $g_{\Theta^{(0)}} = 0$, where $\Theta^{(0)} = (a^{(0)}, B^{(0)}, c^{(0)})$. This is just for theoretical simplicity. Indeed, we can relax the symmetric initialization by considering an additional error stemming from the nonzero initialization in the function space.

**Regularized expected risk minimization.** Instead of minimizing the expected risk (1) itself, we consider the minimization problem of the regularized expected risk around the initial values:

$$\min_{\Theta} \left\{ \mathcal{L}(g_\Theta) + \frac{\lambda}{2} \left( \|a - a^{(0)}\|_2^2 + \|B - B^{(0)}\|_F^2 + \|c - c^{(0)}\|_2^2 \right) \right\}. \tag{3}$$

where the last term is the $L_2$-regularization at an initial point with a regularization parameter $\lambda > 0$. This regularization forces iterations obtained by optimization algorithms to stay close to the initial value, which enables us to utilize the better convergence property of regularized kernel methods.

**Averaged stochastic gradient descent.** Stochastic gradient descent is the most popular method for solving large-scale machine learning problems, and its averaged variant is also frequently used to stabilize and accelerate the convergence. In this study, we analyze the generalization ability of an averaged stochastic gradient descent. The update rule is presented in Algorithm 1. Let $\Theta^{(t)} = (a^{(t)}, B^{(t)}, c^{(t)})$ denote the collection of $t$-th iterates of parameters $a \in \mathbb{R}^M$, $B \in \mathbb{R}^{d \times M}$, and $c \in \mathbb{R}^M$. At $t$-th iterate, stochastic gradient descent using a learning rate $\eta_t$ for the problem (3) with respect to $a, B, c$ is performed on lines 4–6 for a randomly sampled example $(x_t, y_t) \sim \rho$. These updates can be rewritten in an element-wise fashion as follows. For $r \in \{1, \ldots, M\}$,

$$a_r^{(t+1)} - a_r^{(0)} = (1 - \eta_t \lambda)(a_r^{(t)} - a_r^{(0)}) - \eta_t M^{-1/2}(g_{\Theta^{(t)}}(x_t) - y_t)\sigma(b_r^{(t)\top} x_t + \gamma c_r^{(t)}),$$

$$b_r^{(t+1)} - b_r^{(0)} = (1 - \eta_t \lambda)(b_r^{(t)} - b_r^{(0)}) - \eta_t M^{-1/2}(g_{\Theta^{(t)}}(x_t) - y_t)a_r^{(t)}\sigma'(b_r^{(t)\top} x_t + \gamma c_r^{(t)})x_t,$$

$$c_r^{(t+1)} - c_r^{(0)} = (1 - \eta_t \lambda)(c_r^{(t)} - c_r^{(0)}) - \eta_t M^{-1/2}(g_{\Theta^{(t)}}(x_t) - y_t)a_r^{(t)}\gamma\sigma'(b_r^{(t)\top} x_t + \gamma c_r^{(t)}),$$

where $a^{(t)} = (a_1^{(t)}, \ldots, a_M^{(t)})^\top$, $B^{(t)} = (b_1^{(t)}, \ldots, b_M^{(t)})$, and $c^{(t)} = (c_1^{(t)}, \ldots, c_M^{(t)})^\top$. Finally, a weighted average using weights $\alpha_t$ of the history of parameters is computed on line 9. In our theory, we consider the constant learning rate $\eta_t = \eta$ and uniform averaging $\alpha_t = 1/(T+1)$.

---

**Algorithm 1** Averaged Stochastic Gradient Descent

---

1: **Input:** number of iterations $T$, regularization parameter $\lambda$, learning rates $(\eta_t)_{t=0}^{T-1}$, averaging weights $(\alpha_t)_{t=0}^T$, initial values $\Theta^{(0)} = (a^{(0)}, B^{(0)}, c^{(0)})$
2: **for** $t = 0$ **to** $T - 1$ **do**
3:     Randomly draw a sample $(x_t, y_t) \sim \rho$
4:     $a^{(t+1)} \leftarrow a^{(t)} - \eta_t \partial_a \ell(g_{\Theta^{(t)}}(x_t), y_t) - \eta_t \lambda(a^{(t)} - a^{(0)})$
5:     $B^{(t+1)} \leftarrow B^{(t)} - \eta_t \partial_B \ell(g_{\Theta^{(t)}}(x_t), y_t) - \eta_t \lambda(B^{(t)} - B^{(0)})$
6:     $c^{(t+1)} \leftarrow c^{(t)} - \eta_t \partial_c \ell(g_{\Theta^{(t)}}(x_t), y_t) - \eta_t \lambda(c^{(t)} - c^{(0)})$
7:     $\Theta^{(t+1)} \leftarrow (a^{(t+1)}, B^{(t+1)}, c^{(t+1)})$
8: **end for**
9: $\overline{\Theta}^{(T)} = (\sum_{t=0}^T \alpha_t a^{(t)}, \sum_{t=0}^T \alpha_t B^{(t)}, \sum_{t=0}^T \alpha_t c^{(t)})$
10: Return $g_{\overline{\Theta}^{(T)}}$

---

**Integral and Covariance Operators.** The integral and covariance operators associated with the kernels, which are the limit of the Gram-matrix as the number of examples goes to infinity, play a crucial role in determining the learning speed. For a given Hilbert space $\mathcal{H}$, we denote by $\otimes_{\mathcal{H}}$ the tensor product on $\mathcal{H}$, that is, $\forall (f, g) \in \mathcal{H}^2$, $f \otimes_{\mathcal{H}} g$ defines a linear operator; $h \in \mathcal{H} \mapsto (f \otimes_{\mathcal{H}} g)h = \langle f, h \rangle_{\mathcal{H}} g \in \mathcal{H}$. Note that $f \otimes_{\mathcal{H}} g$ naturally induces a bilinear function: $(h, h') \in \mathcal{H} \times \mathcal{H} \mapsto \langle (f \otimes_{\mathcal{H}} g)h, h' \rangle_{\mathcal{H}} = \langle f, h \rangle_{\mathcal{H}} \langle g, h' \rangle_{\mathcal{H}}$. When $\mathcal{H}$ is a reproducing kernel Hilbert space (RKHS) associated with a bounded kernel $k : \mathcal{X} \times \mathcal{X} \to \mathbb{R}$, the *covariance operator* $\Sigma : \mathcal{H} \mapsto \mathcal{H}$ is defined as follows: Set $K_X \overset{def}{=} k(X, \cdot)$ and

$$\Sigma = \mathbb{E}_{X \sim \rho_X}[K_X \otimes_{\mathcal{H}} K_X].$$

Note that the covariance operator is a restriction of the *integral operator* on $L_2(\rho_X)$:

$$f \in L_2(\rho_X) \longmapsto \Sigma f = \int_{\mathcal{X}} f(X) K_X \mathrm{d}\rho_X \in L_2(\rho_X).$$

We use the same symbol as above for convenience with a slight abuse of notation. Because $\Sigma$ is a compact self-adjoint operator on $L_2(\rho_X)$, $\Sigma$ has the following eigendecomposition: $\Sigma f = \sum_{i=1}^{\infty} \lambda_i \langle f, \phi_i \rangle_{L_2(\rho_X)} \phi_i$ for $f \in L_2(\rho_X)$, where $\{(\lambda_i, \phi_i)\}_{i=1}^{\infty}$ is a pair of eigenvalues and orthogonal eigenfunctions in $L_2(\rho_X)$. For $s \in \mathbb{R}$, the power $\Sigma^s$ is defined as $\Sigma^s f = \sum_{i=1}^{\infty} \lambda_i^s \langle f, \phi_i \rangle_{L_2(\rho_X)} \phi_i$.

## 3 MAIN RESULTS: MINIMAX OPTIMAL CONVERGENCE RATES

In this section, we present the main results regarding the fast convergence rates of the averaged stochastic gradient descent under a certain condition on the NTK and target function $g_\rho$.

**Neural tangent kernel.** The NTK is a recently developed kernel function and has been shown to be extremely useful in demonstrating the global convergence of the gradient descent method for neural networks (cf., Jacot et al. (2018); Chizat & Bach (2018b); Du et al. (2019b); Allen-Zhu et al. (2019a;b); Arora et al. (2019a)). The NTK in our setting is defined as follows: $\forall x, \forall x' \in \mathcal{X}$,

$$k_\infty(x, x') \stackrel{def}{=} \mathbb{E}_{b^{(0)}}[\sigma(b^{(0)\top} x) \sigma(b^{(0)\top} x')] + R^2(x^\top x' + \gamma^2) \mathbb{E}_{b^{(0)}}[\sigma'(b^{(0)\top} x) \sigma'(b^{(0)\top} x')], \quad (4)$$

where the expectation is taken with respect to $b^{(0)} \sim \mu_0$. The NTK is the key to the global convergence of a neural network because it makes a connection between the (averaged) stochastic gradient descent for a neural network and the RKHS associated with $k_\infty$ (see Proposition A). Although this type of connection has been shown in previous studies (Arora et al., 2019b; Weinan et al., 2019; Lee et al., 2019; 2020), note that their results are inapplicable to our theory because we consider the population risk. Indeed, our study is the first to establish this connection for an (averaged) stochastic gradient descent in terms of the uniform distance on the support of the data distribution, enabling us to obtain faster convergence rates. We note that an NTK $k_\infty$ is the sum of two NTKs, that is, the first and second terms in (4) are NTKs for the output and input layers with bias, respectively.

### 3.1 GLOBAL CONVERGENCE ANALYSIS

Let $\mathcal{H}_\infty$ be an RKHS associated with NTK $k_\infty$, and let $\Sigma_\infty$ be the corresponding integral operator. Let $\{\lambda_i\}_{i=1}^{\infty}$ denote the eigenvalues of $\Sigma_\infty$ sorted in decreasing order: $\lambda_1 \geq \lambda_2 \geq \cdots$.

**Assumption 1.**

**(A1)** *There exists $C > 0$ such that $\|\sigma''\|_\infty \leq C$, $\|\sigma'\|_\infty \leq 2$, and $|\sigma(u)| \leq 1 + |u|$ for $\forall u \in \mathbb{R}$.*

**(A2)** $\mathrm{supp}(\rho_X) \subset \{x \in \mathbb{R}^d \mid \|x\|_2 \leq 1\}$, $\mathcal{Y} \subset [-1, 1]$, $R = 1$, *and* $\gamma \in [0, 1]$.

**(A3)** *There exists $r \in [1/2, 1]$ such that $g_\rho \in \Sigma_\infty^r(L_2(\rho_X))$, i.e., $\|\Sigma_\infty^{-r} g_\rho\|_{L_2(\rho_X)} < \infty$.*

**(A4)** *There exists $\beta > 1$ such that $\lambda_i = \Theta(i^{-\beta})$.*

**Remark.**

- **(A1):** Typical smooth activation functions, such as sigmoid and tanh functions, and smooth approximations of the ReLU, such as swish (Ramachandran et al., 2017), which performs as well as or even better than the ReLU, satisfy Assumption **(A1)**. This condition is used to relate the two learning dynamics between neural networks and kernel methods (see Proposition A).

- **(A2):** The boundedness **(A2)** of the feature space and label are often assumed for stochastic optimization and least squares regression for theoretical guarantees (see Steinwart et al. (2009)). Note that these constants in **(A2)** can be relaxed to arbitrary constants.

- **(A3):** Assumption **(A3)** measures the complexity of $g_\rho$ because $\Sigma_\infty$ can be considered as a smoothing operator using a kernel $k_\infty$. A larger $r$ indicates a faster decay of the coefficients of expansion of $g_\rho$ based on the eigenfunctions of $\Sigma_\infty$ and smoothens $g_\rho$. In addition, $\Sigma_\infty^r(L_2(\rho_X))$

shrinks with respect to $r$ and $\Sigma_\infty^{1/2}(L_2(\rho_X)) = \mathcal{H}_\infty$, resulting in $g_\rho \in \mathcal{H}_\infty$. This condition is used to control the bias of the estimators through $L_2$-regularization. The notation $\Sigma_\infty^{-r} g_\rho$ represents any function $G \in L_2(\rho_X)$ such that $g_\rho = \Sigma_\infty^r G$.

- **(A4):** Assumption **(A4)** controls the complexity of the hypothesis class $\mathcal{H}_\infty$. A larger $\beta$ indicates a faster decay of the eigenvalues and makes $\mathcal{H}_\infty$ smaller. This assumption is essentially needed to bound the variance of the estimators efficiently and derive a fast convergence rate. Theorem 1 and Corollary 1, 2 hold even though the condition in **(A4)** is relaxed to $\lambda_i = O(i^{-\beta})$ and the lower bound $\lambda_i = \Omega(i^{-\beta})$ is necessary only for making obtained rates minimax optimal.

Under these assumptions, we derive the convergence rate of the averaged stochastic gradient descent for an overparameterized two-layer neural network, the proof is provided in the Appendix.

**Theorem 1.** *Suppose Assumptions **(A1)-(A3)** hold. Run Algorithm 1 with a constant learning rate $\eta$ satisfying $4(6 + \lambda)\eta \leq 1$. Then, for any $\epsilon > 0$, $\|\Sigma_\infty\|_{\mathrm{op}} \geq \lambda > 0$, $\delta \in (0, 1)$, and $T \in \mathbb{Z}_+$, there exists $M_0 \in \mathbb{Z}_+$ such that for any $M \geq M_0$, the following holds with high probability at least $1 - \delta$ over the random choice of features $\Theta^{(0)}$:*

$$\mathbb{E}\left[\|g_{\overline{\Theta}^{(T)}} - g_\rho\|_{L_2(\rho_X)}^2\right] \leq \epsilon + \alpha\left(\lambda^{2r}\|\Sigma_\infty^{-r} g_\rho\|_{L_2(\rho_X)}^2 + \frac{1}{T+1}\|g_\rho\|_{\mathcal{H}_\infty}^2 + \frac{1}{\lambda\eta^2(T+1)^2}\|g_\rho\|_{\mathcal{H}_\infty}^2\right)$$
$$+ \frac{\alpha}{T+1}\left(1 + \|g_\rho\|_{L_2(\rho_X)}^2 + \|\Sigma_\infty^{-r} g_\rho\|_{L_2(\rho_X)}^2\right)\mathrm{Tr}\left(\Sigma_\infty(\Sigma_\infty + \lambda I)^{-1}\right),$$

*where $\alpha > 0$ is a universal constant and $g_{\overline{\Theta}^{(T)}}$ is an iterate obtained through Algorithm 1.*

**Remark.** The first term $\epsilon$ and second term $\lambda^{2r}\|\Sigma_\infty^{-r} g_\rho\|_{L_2(\rho_X)}^2$ are the approximation error and bias, which can be chosen to be arbitrarily small. The first term comes from the approximation of the NTK using finite-sized neural networks, and the second term comes from the $L_2$-regularization, which coincides with a bias term in the theory of least squares regression (Caponnetto & De Vito, 2007). The third and fourth terms come from the convergence of the averaged semi-stochastic gradient descent (which is considered in the proof) in terms of the optimization. The appearance of an inverse dependence on $\lambda$ in the fourth term is common because a smaller $\lambda$ indicates a weaker strong convexity, which slows down the convergence speed of the optimization methods (Rakhlin et al., 2012). The term $\mathrm{Tr}\left(\Sigma_\infty(\Sigma_\infty + \lambda I)^{-1}\right)$ is the variance from the stochastic approximation of the gradient, and it is referred to as the *degree of freedom* or the *effective dimension*, which is known to be unavoidable in kernel regression problems (Caponnetto & De Vito, 2007; Dieuleveut & Bach, 2016; Rudi & Rosasco, 2017).

**Global convergence in NTK regime.** This theorem shows the global convergence to the Bayes rule $g_\rho$, which is a minimizer over all measurable maps because the approximation term $\epsilon$ can be arbitrarily small by taking a sufficiently large network width $M$. The required value of $M$ has an exponential dependence on $T$; note, however, that reducing $M$ is not the main focus of the present study. The key technique is to relate two learning dynamics for two-layer neural networks and kernel methods in an RKHS approximating $\mathcal{H}_\infty$ up to a small error. Unlike existing studies (Du et al., 2019b; Arora et al., 2019a;b; Weinan et al., 2019; Lee et al., 2019; 2020) showing such connections, we establish this connection in term of the $L_\infty(\rho_X)$-norm, which is more useful in a generalization analysis. Moreover, existing studies essentially rely on the strict positivity of the Gram-matrix to localize all iterates around an initial value, which can slow down the convergence rate in terms of the generalization because the convergence of the eigenvalues of the NTK to zero affects the Rademacher complexity. By contrast, our theory succeeds in demonstrating the global convergence in the NTK regime without the positivity of the NTK.

## 3.2 OPTIMAL CONVERGENCE RATE

We derive the fast convergence rate from Theorem 1 by utilizing Assumption **(A4)**, which defines the complexity of the NTK. The regularization parameter $\lambda$ mainly controls the trade-off within the generalization bound, that is, a smaller value decreases the bias term but increases the variance term including the degree of freedom. The degree of freedom $\mathrm{Tr}\left(\Sigma_\infty(\Sigma_\infty + \lambda I)^{-1}\right)$ can be specified by imposing Assumption **(A4)** because it determines the decay rate of the eigenvalues of $\Sigma_\infty$. As a result, this trade-off between bias and variance depending on the choice of $\lambda$ becomes clear, and we

can determine the optimal value. Concretely, by setting $\lambda = T^{-\beta/(2r\beta+1)}$, the sum of the bias and variance terms is minimized, and these terms become asymptotically equivalent.

**Corollary 1.** *Suppose Assumptions* **(A1)-(A4)** *hold. Run Algorithm 1 with the constant learning rate* $\eta = O(1)$ *satisfying* $4(6 + \lambda)\eta \leq 1$ *and* $\lambda = T^{-\beta/(2r\beta+1)}$. *Then, for any* $\epsilon > 0$, $\delta \in (0, 1)$ *and* $T \in \mathbb{Z}_+$ *satisfying* $\|\Sigma_\infty\|_{op} \geq \lambda$, *there exists* $M_0 \in \mathbb{Z}_+$ *such that for any* $M \geq M_0$, *the following holds with high probability at least* $1 - \delta$ *over the random choice of random features* $\Theta^{(0)}$:

$$\mathbb{E}\left[\|g_{\overline{\Theta}^{(T)}} - g_\rho\|^2_{L_2(\rho_X)}\right] \leq \epsilon + \alpha T^{\frac{-2r\beta}{2r\beta+1}} \left(1 + \|\Sigma_\infty^{-r} g_\rho\|^2_{L_2(\rho_X)}\right),$$

*where* $\alpha > 0$ *is a universal constant and* $g_{\overline{\Theta}^{(T)}}$ *is an iterate obtained by Algorithm 1.*

The resulting convergence rate is $O(T^{\frac{-2r\beta}{2r\beta+1}})$ with respect to $T$ by considering a sufficiently large network width of $M$ such that the error $\epsilon$ stemming from the approximation of NTK can be ignored. Because $T$ corresponds to the number of examples used to learn a predictor $g_{\overline{\Theta}^{(T)}}$, this convergence rate is simply the generalization error bound for the averaged stochastic gradient descent. In general, this rate is always faster than $T^{-1/2}$ and is known to be the minimax optimal rate of estimation (Caponnetto & De Vito, 2007; Blanchard & Mücke, 2018) in $\mathcal{H}_\infty$ in the following sense. Let $\mathcal{P}(\beta, r)$ be a data distribution class satisfying Assumptions **(A2)-(A4)**. Then,

$$\lim_{\tau \to 0} \liminf_{T \to \infty} \inf_{h^{(T)}} \sup_\rho \mathbb{P}\left[\|h^{(T)} - g_\rho\|^2_{L_2(\rho_X)} > \tau T^{\frac{-2r\beta}{2r\beta+1}}\right] = 1,$$

where $\rho$ is taken in $\mathcal{P}(\beta, r)$ and $h^{(T)}$ is taken over all mappings $(x_t, y_t)_{t=0}^{T-1} \mapsto h^{(T)} \in \mathcal{H}_\infty$.

## 3.3 Explicit Optimal Convergence Rate for Smooth Approximation of ReLU

For smooth activation functions that sufficiently approximate the ReLU, an optimal explicit convergence rate can be derived under the setting in which the target function is specified by NTK with the ReLU, and the data are distributed uniformly on a sphere. We denote the ReLU activation by $\sigma(u) = \max\{0, u\}$ and a smooth approximation of ReLU by $\sigma^{(s)}$, which converges to ReLU, as $s \to \infty$ in the following sense. We make alternative assumptions to **(A1)**, **(A2)**, and **(A3)**:

**Assumption 2.**

**(A1')** $\sigma^{(s)}$ *satisfies* **(A1)**. $\sigma^{(s)}$ *and* $\sigma^{(s)'}$ *converge pointwise almost surely to* $\sigma$ *and* $\sigma'$ *as* $s \to \infty$.

**(A2')** $\rho_X$ *is a uniform distribution on* $\{x \in \mathbb{R}^d \mid \|x\|_2 = 1\}$. $\mathcal{Y} \subset [-1, 1]$, $R = 1$, *and* $\gamma \in (0, 1]$.

**(A3')** *The condition* **(A3)** *is satisfied by the NTK associated with the ReLU activation* $\sigma$.

**(A1')** and **(A2')** are special cases of **(A1)** and **(A2)**. There are several activation functions that satisfy this condition, including swish (Ramachandran et al., 2017): $\sigma^{(s)}(u) = \frac{u}{1+\exp(-su)}$. Under these conditions, we can estimate the decay rate of the eigenvalues for the ReLU as $\beta = 1 + \frac{1}{d-1}$, yielding the explicit optimal convergence rate by adapting the proof of Theorem 1 to the current setting. Note that Algorithm 1 is run for a neural network with a smooth approximation $\sigma^{(s)}$ of the ReLU.

**Corollary 2.** *Suppose Assumptions* **(A1')**, **(A2')**, *and* **(A3')** *hold. Run Algorithm 1 with the constant learning rate* $\eta = O(1)$ *satisfying* $4(6 + \lambda)\eta \leq 1$, *and* $\lambda = T^{-d/(2rd+d-1)}$. *Given any* $\epsilon > 0$, $\delta \in (0, 1)$ *and* $T \in \mathbb{Z}_+$ *satisfying* $\|\Sigma_\infty\|_{op} \geq 2\lambda$, *let* $s$ *be an arbitrary and sufficiently large positive value. Then, there exists* $M_0 \in \mathbb{Z}_+$ *such that for any* $M \geq M_0$, *the following holds with high probability at least* $1 - \delta$ *over the random choice of random features* $\Theta^{(0)}$:

$$\mathbb{E}\left[\|g_{\overline{\Theta}^{(T)}} - g_\rho\|^2_{L_2(\rho_X)}\right] \leq \epsilon + \alpha T^{\frac{-2rd}{2rd+d-1}} \left(1 + \|\Sigma_\infty^{-r} g_\rho\|^2_{L_2(\rho_X)}\right),$$

*where* $\alpha > 0$ *is a universal constant and* $g_{\overline{\Theta}^{(T)}}$ *is an iterate obtained by Algorithm 1.*

## 4 Experiments

We verify the importance of the specification of target functions by showing the misspecification significantly slows down the convergence speed. To evaluate the misspecification, we consider

single-layer learning as well as the two-layer learning, and we see the advantage of two-layer learning. Here, note that, with evident modification of the proofs, the counterparts of Corollaries 1 and 2 for learning a single layer also hold by replacing $\Sigma_\infty$ with the covariance operator $\Sigma_{a,\infty}$ ($\Sigma_{b,\infty}$) associated with $k_{a,\infty}$ ($k_{b,\infty}$), where

$$k_{a,\infty}(x,x') = \mathbb{E}_{b^{(0)}}[\sigma(b^{(0)\top}x)\sigma(b^{(0)\top}x')],$$
$$k_{b,\infty}(x,x') = R^2(x^\top x' + \gamma^2)\mathbb{E}_{b^{(0)}}[\sigma'(b^{(0)\top}x)\sigma'(b^{(0)\top}x')],$$

which are components of $k_\infty = k_{a,\infty} + k_{b,\infty}$ corresponding to the output and input layers, respectively. Then, from Corollaries 1 and 2, a Bayes rule $g_\rho$ is learned efficiently by optimizing the layer which has a small norm $\|\Sigma^{-r}g_\rho\|_{L_2(\rho_X)}$ for $\Sigma \in \{\Sigma_{a,\infty}, \Sigma_{b,\infty}, \Sigma_\infty\}$.

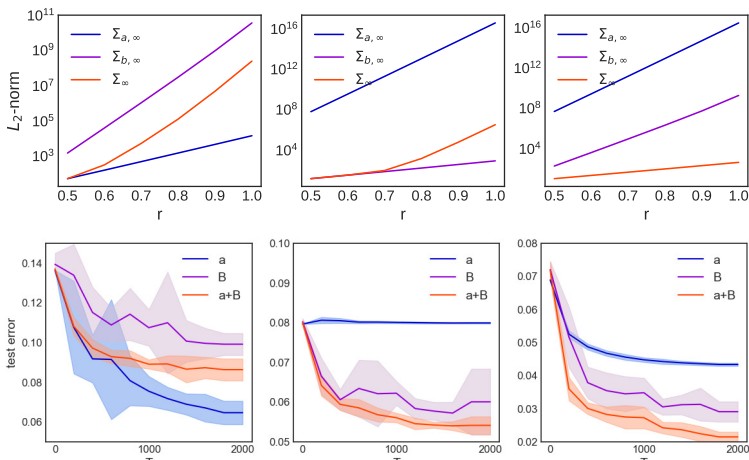

Figure 2: Top: Estimation of $\|\Sigma^{-r}g_\rho\|_{L_2(\rho_X)}$ ($r \in [0.5, 1]$) for integral operators $\Sigma \in \{\Sigma_{a,\infty}, \Sigma_{b,\infty}, \Sigma_\infty\}$ of two-layer ReLU networks. Bayes rules $g_\rho$ are set to the average eigenfunctions of $\Sigma_{a,\infty}$ (left), $\Sigma_{b,\infty}$ (middle), and $\Sigma_\infty$ (right). Bottom: Learning curves of test errors for Algorithm 1 with two-layer swish networks.

**Experimental settings.** Figure 2 (Top) depicts norms $\|\Sigma^{-r}g_\rho\|_{L_2(\rho_X)}$ for $\Sigma \in \{\Sigma_{a,\infty}, \Sigma_{b,\infty}, \Sigma_\infty\}$. Bayes rules $g_\rho$ are averages of eigenfunctions of $\Sigma_{a,\infty}$ (left), $\Sigma_{b,\infty}$ (middle), and $\Sigma_\infty$ (right) corresponding to the 10-largest eigenvalues excluding the first and second, with the setting: $R = 1/(20\sqrt{2})$, $\gamma = 10\sqrt{2}$, and $\rho_X$ is the uniform distribution on the unit sphere in $\mathbb{R}^2$. To estimate eigenvalues and eigenfunctions, we draw $10^4$-samples from $\rho_X$ and $M = 2 \times 10^4$-hidden nodes of a two-layer ReLU.

**Empirical observations.** We observe $g_\rho$ has the smallest norm with respect to the integral operator which specifies $g_\rho$ and has a comparably small norm with respect to $\Sigma_\infty$ even for the cases where $g_\rho$ is specified by $\Sigma_{a,\infty}$ or $\Sigma_{b,\infty}$. This observation suggests the efficiency of learning a corresponding layer to $g_\rho$ and learning both layers, and it is empirically verified. We run Algorithm 1 10-times with respect to output (blue), input (purple), and both layers (orange) of two-layer swish networks with $s = 10$. Figure 2 (Bottom) depicts the average and standard deviation of test errors. From the figure, we see that learning a corresponding layer to $g_\rho$ and both layers exhibit faster convergence, and that misspecification significantly slows down the convergence speed in all cases.

## 5 CONCLUSION

We analyzed the convergence of the averaged stochastic gradient descent for overparameterized two-layer neural networks for a regression problem. Through the development of a new proof strategy that does not rely on the positivity of the NTK, we proved that the global convergence (Theorem 1) relies only on the overparameterization. Moreover, we demonstrated the minimax optimal convergence rates (Corollary 1) in terms of the generalization error depending on the complexities of the target function and the hypothesis class and showed the explicit optimal rate for the smooth approximation of the ReLU.

ACKNOWLEDGMENTS

AN was partially supported by JSPS Kakenhi (19K20337) and JST-PRESTO. TS was partially supported by JSPS KAKENHI (18K19793, 18H03201, and 20H00576), Japan Digital Design, and JST CREST.

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

# Appendix

## A  PROOF SKETCH OF THE MAIN RESULTS

We provide several key results and a proof sketch of Theorem 1 and Corollary 1. We first recall the definition of stochastic gradients of $\mathcal{L}$ in a general RKHS $(\mathcal{H}, \langle, \rangle_{\mathcal{H}})$ associated with a uniformly bounded real-valued kernel function $k : \mathcal{X} \times \mathcal{X} \to \mathbb{R}$. We set $K_X = k(X, \cdot)$. Then, it follows that for $\forall g, \forall h \in \mathcal{H}$,

$$\mathcal{L}(g + h) = \mathcal{L}(g) + \langle \mathbb{E}[\partial_z \ell(g(X), Y) K_X], h \rangle_{\mathcal{H}} + o(\|h\|_{\mathcal{H}}),$$

which is confirmed by the following equations:

$$\mathbb{E}[l((g + h)(X), Y)] = \mathbb{E}[l(g(X), Y) + \partial_\zeta l(g(X), Y) h(X) + o(|h(X)|)],$$

$h(X) = \langle h, k(X, \cdot) \rangle_{\mathcal{H}}$, and $|h(X)| \leq \|h\|_{\mathcal{H}} \sqrt{k(X, X)}$. This means that the stochastic gradient of $\mathcal{L}$ in $\mathcal{H}$ is given by $\partial_\zeta \ell(g(X), Y) k(X, \cdot)$ for $(X, Y) \sim \rho$. In addition, the stochastic gradient of the $L_2$-regularized risk is given by $\partial_\zeta \ell(g(X), Y) k(X, \cdot) + \lambda g$.

### A. 1  REFERENCE AVERAGED STOCHASTIC GRADIENT DESCENT

We consider a *random feature approximation* of NTK $k_\infty$: for an initial value $B^{(0)} = (b_r^{(0)})_{r=1}^M$, $\forall x, \forall x' \in \mathcal{X}$,

$$k_M(x, x') \overset{def}{=} \frac{1}{M} \sum_{r=1}^M \sigma(b_r^{(0)\top} x) \sigma(b_r^{(0)\top} x') + \frac{(x^\top x' + \gamma^2)}{M} \sum_{r=1}^M \sigma'(b_r^{(0)\top} x) \sigma'(b_r^{(0)\top} x'), \quad (5)$$

We can confirm that $k_M$ is an approximation of NTK, that is, $k_M$ converges to $k_\infty$ uniformly over $\mathrm{supp}(\rho_X) \times \mathrm{supp}(\rho_X)$ almost surely by the uniform law of large numbers. We denote by $(\mathcal{H}_M, \langle, \rangle_{\mathcal{H}_M})$ an RKHS associated with $k_M$. By the assumptions, we see $k_M(x, x') \leq 12$ for $\forall (x, x') \in \mathrm{supp}(\rho_X) \times \mathrm{supp}(\rho_X)$.

We introduce averaged stochastic gradient descent in $\mathcal{H}_M$ (see Algorithm 2) as a reference for Algorithm 1. The notation $G^{(t)}$ represents a stochastic gradient at the $t$-th iterate:

$$G^{(t)} \overset{def}{=} \partial_z \ell(g^{(t)}(x_t), y_t) k_M(x_t, \cdot).$$

---

**Algorithm 2** Reference ASGD in $\mathcal{H}_M$

---

1: **Input:** number of iterations $T$, regularization parameter $\lambda$, learning rates $(\eta_t)_{0=1}^{T-1}$, averaging weights $(\alpha_t)_{t=0}^T$,
2: $g^{(0)} \leftarrow 0$
3: **for** $t = 0$ **to** $T - 1$ **do**
4:    Randomly draw a sample $(x_t, y_t) \sim \rho$
5:    $g^{(t+1)} \leftarrow (1 - \eta_t \lambda) g^{(t)} - \eta_t G^{(t)}$
6: **end for**
7: Return $\overline{g}^{(T)} = \sum_{t=0}^T \alpha_t g^{(t)}$

---

The following proposition shows the equivalence between the averaged stochastic gradient descent for two-layer neural networks and that in $\mathcal{H}_M$ up to a small constant depending on $M$.

**Proposition A.** *Suppose Assumptions* **(A1)** *and* **(A2)** *hold. Run Algorithms 1 and 2 with the constant learning rate $\eta_t = \eta$ satisfying $\eta \lambda < 1$ and $\eta \leq 1$. Moreover, assume that they share the same hyper-parameter settings and the same examples $(x_t, y_t)_{t=0}^{T-1}$ to compute stochastic gradient. Then, for arbitrary $T \in \mathbb{Z}_+$ and $\epsilon > 0$, there exists $M \in \mathbb{Z}_+$ depending only on $T$ and $\epsilon$ such that $\forall t \leq T$,*

$$\|g_{\overline{\Theta}^{(t)}} - \overline{g}^{(t)}\|_{L_\infty(\rho_X)} \leq \epsilon, \quad (6)$$

*where $g_{\overline{\Theta}^{(t)}}$ and $\overline{g}^{(t)}$ are iterates obtained by Algorithm 1 and 2, respectively.*

**Remark.** Note that this proposition holds for non-averaged SGD too because it is a special case of averaged SGD by setting only one $\alpha_t$ to 1.

**Key idea.** This proposition is the key because it connects two learning dynamics for neural networks and RKHS $\mathcal{H}_M$ by utilizing overparameterization without the positivity of NTK unlike existing studies (Weinan et al., 2019; Arora et al., 2019b) that provide such a connection for continuous gradient flow with the positive NTK. Instead of the positivity of NTK, Proposition A says that overparameterization increases the time stayed in the NTK regime where the learning dynamics for neural networks can be characterized by the NTK. As a result, because $M$ is free from the other hyper-parameters, the averaged stochastic gradient descent for the overparameterized two-layer neural networks can fully inherit preferable properties from learning dynamics in $\mathcal{H}_M$ with an appropriate choice of learning rates and regularization parameters as long as the network width is sufficiently large depending only on the number of iterations and the required accuracy.

## A. 2 Convergence Rate of the Reference ASGD

We give the convergence analysis of Algorithm 2 in $\mathcal{H}_M$, which will be a part of a bound in Theorem 1. Proofs essentially rely on several techniques developed in serial studies (Bach & Moulines, 2013; Dieuleveut & Bach, 2016; Dieuleveut et al., 2017; Pillaud-Vivien et al., 2018a; Rudi & Rosasco, 2017; Carratino et al., 2018) with several adaptations to our settings.

Let $M \in \mathbb{Z}_+ \cup \{\infty\}$ be a positive number or $\infty$. We set $K_{M,X} \stackrel{def}{=} k_M(X, \cdot)$ and denote by $\Sigma_M$ the covariance operator defined by $k_M$:

$$\Sigma_M \stackrel{def}{=} \mathbb{E}_{X \sim \rho_X}[K_{M,X} \otimes_{\mathcal{H}_M} K_{M,X}].$$

We denote by $g_{M,\lambda}$ the minimizer of the regularized risk over $\mathcal{H}_M$:

$$g_{M,\lambda} \stackrel{def}{=} \arg\min_{g \in \mathcal{H}_M} \left\{ \mathcal{L}(g) + \frac{\lambda}{2}\|g\|_{\mathcal{H}_M}^2 \right\}.$$

We remark that $\Sigma_M : L_2(\rho_X) \to \mathcal{H}_M$ is isometric (Cucker & Smale, 2002), that is, $\forall (f, g) \in L_2(\rho_X) \times L_2(\rho_X)$,

$$\left\langle \Sigma_M^{1/2} f, \Sigma_M^{1/2} g \right\rangle_{\mathcal{H}_M} = \langle f, g \rangle_{L_2(\rho_X)},$$

and we use this fact frequently. It is known that $g_{M,\lambda}$ is represented as follows (Caponnetto & De Vito, 2007):

$$\begin{aligned} g_{M,\lambda} &= (\Sigma_M + \lambda I)^{-1} \mathbb{E}_{(X,Y)}[Y K_{M,X}] \\ &= (\Sigma_M + \lambda I)^{-1} \Sigma_M g_\rho. \end{aligned} \tag{7}$$

The following theorem provides a convergence rate of Algorithm 2 to the minimizer $g_{M,\lambda}$.

**Theorem A.** *Suppose Assumptions* **(A1)***,* **(A2)** *and* **(A3)** *hold. Run Algorithm 2 with the constant learning rate $\eta_t = \eta$ satisfying $4(6 + \lambda)\eta \leq 1$. Then, for $\forall \lambda > 0$ and $\forall \delta \in (0, 1)$ there exists $M_0 > 0$ such that for $\forall M \geq M_0$ the following holds with high probability at least $1 - \delta$:*

$$\begin{aligned} \mathbb{E}\left[\left\|\overline{g}^{(T)} - g_{M,\lambda}\right\|_{L_2(\rho_X)}^2\right] &\leq \frac{4}{\eta^2(T+1)^2}\|(\Sigma_M + \lambda I)^{-1}g_{M,\lambda}\|_{L_2(\rho_X)}^2 \\ &\quad + \frac{2 \cdot 24^2}{T+1}\|(\Sigma_M + \lambda I)^{-1/2}g_{M,\lambda}\|_{L_2(\rho_X)}^2 \\ &\quad + \frac{8}{T+1}\left(1 + \|g_\rho\|_{L_2(\rho_X)}^2 + 24\|\Sigma_\infty^{-r}g_\rho\|_{L_2(\rho_X)}^2\right) \mathrm{Tr}\left(\Sigma_M(\Sigma_M + \lambda I)^{-1}\right), \end{aligned}$$

*where $\overline{g}^{(T)}$ is an iterate obtained by Algorithm 2.*

**Remark.** The first and second terms stem from the optimization speed of a semi-stochastic part of averaged stochastic gradient descent. The first term has a better dependency on $T$, but it has a worse dependency on $\lambda$ than the second one. This kind of deterioration due to the weak strong convexity is common in first-order optimization methods. However, as confirmed later, these two

terms are dominated by the variance term corresponding to the third term by setting hyper-parameters appropriately.

To make the bound in Theorem A free from the size of $M$, we introduce the following proposition.

**Proposition B.** *Suppose $g_\rho \in \mathcal{H}_\infty$ holds. Under Assumption* **(A1)** *and* **(A2)***, for any $\delta \in (0, 1)$, there exists $M_0 \in \mathbb{Z}_+$ such that for any $M \geq M_0$, the following holds with high probability at least $1 - \delta$:*

$$\|(\Sigma_M + \lambda I)^{-1} g_{M,\lambda}\|_{L_2(\rho_X)}^2 \leq 2\lambda^{-1} \|g_\rho\|_{\mathcal{H}_\infty}^2,$$

$$\|(\Sigma_M + \lambda I)^{-1/2} g_{M,\lambda}\|_{L_2(\rho_X)}^2 \leq 2\|g_\rho\|_{\mathcal{H}_\infty}^2,$$

*and if $\lambda \leq \|\Sigma_\infty\|_{\mathrm{op}}$, then*

$$\mathrm{Tr}\left(\Sigma_M(\Sigma_M + \lambda I)^{-1}\right) \leq 3\mathrm{Tr}\left(\Sigma_\infty(\Sigma_\infty + \lambda I)^{-1}\right).$$

**Remark.** The last inequality on the degree of freedom was shown in Rudi & Rosasco (2017).

To show the convergence to $g_\rho$, we utilize the following decomposition:

$$\frac{1}{3}\|\overline{g}^{(T)} - g_\rho\|_{L_2(\rho_X)}^2 \leq \|\overline{g}^{(T)} - g_{M,\lambda}\|_{L_2(\rho_X)}^2 + \|g_{M,\lambda} - g_{\infty,\lambda}\|_{L_2(\rho_X)}^2 + \|g_{\infty,\lambda} - g_\rho\|_{L_2(\rho_X)}^2,$$

(8)

where $g_{\infty,\lambda} \overset{def}{=} \arg\min_{g \in \mathcal{H}_\infty}\{\mathcal{L}(g) + \frac{\lambda}{2}\|g\|_{\mathcal{H}_\infty}^2\}$.

The first term is the optimization speed evaluated in Theorem A, and the second and third terms are approximation errors from a random feature approximation of NTK and imposing $L_2$-regularization, respectively. These approximation terms can be evaluated by the following existing results. The next proposition is a simplified version of Lemma 8 in Carratino et al. (2018)

**Proposition C** (Carratino et al. (2018))**.** *Under Assumption* **(A1)**, **(A2)***, and* **(A3)***, for any $\epsilon, \lambda > 0$ and $\delta \in (0, 1]$, there exists $M_0 \in \mathbb{Z}_+$ depending on $\epsilon, \lambda, \delta$ such that for any $M \geq M_0$, the following holds with high probability at least $1 - \delta$:*

$$\|g_{M,\lambda} - g_{\infty,\lambda}\|_{L_2(\rho_X)}^2 \leq \epsilon.$$

**Proposition D** (Caponnetto & De Vito (2007))**.** *Under Assumption* **(A3)***, it follows that*

$$\|g_{\infty,\lambda} - g_\rho\|_{L_2(\rho_X)}^2 \leq \lambda^{2r}\|\Sigma_\infty^{-r} g_\rho\|_{L_2(\rho_X)}^2.$$

By combining Theorem A, Proposition B, C, and D with the decomposition (8), we can establish the convergence rate of reference ASGD to reach $g_\rho$, which is simply the generalization error bound.

**Theorem B.** *Assume the same conditions as in Theorem A. Then, for $\forall \epsilon > 0$, $\|\Sigma_\infty\|_{\mathrm{op}} \geq \forall \lambda > 0$, and $\forall \delta \in (0, 1)$, there exists $M_0 \in \mathbb{Z}_+$ such that for $\forall M \geq M_0$, the following holds with high probability at least $1 - \delta$ over the random choice of random features $\Theta^{(0)}$:*

$$\mathbb{E}\left[\|\overline{g}^{(T)} - g_\rho\|_{L_2(\rho_X)}^2\right] \leq \epsilon + 3\lambda^{2r}\|\Sigma_\infty^{-r} g_\rho\|_{L_2(\rho_X)}^2$$

$$+ \frac{24}{T+1}\left(288 + \frac{1}{\lambda\eta^2(T+1)}\right)\|g_\rho\|_{\mathcal{H}_\infty}^2$$

$$+ \frac{24}{T+1}\left(1 + \|g_\rho\|_{L_2(\rho_X)}^2 + 24\|\Sigma_\infty^{-r} g_\rho\|_{L_2(\rho_X)}^2\right)\mathrm{Tr}\left(\Sigma_\infty(\Sigma_\infty + \lambda I)^{-1}\right),$$

*where $\overline{g}^{(T)}$ is an iterate obtained by Algorithm 2.*

### A.3 CONVERGENCE RATES OF ASGD FOR NEURAL NETWORKS

As explained earlier, the generalization bound for the reference ASGD is inherited by that for two-layer neural networks through Proposition A with the following decomposition: for an iterate $\overline{\Theta}_T$ obtained by Algorithm 1,

$$\|g_{\overline{\Theta}^{(T)}} - g_\rho\|_{L_2(\rho_X)} \leq \|g_{\overline{\Theta}^{(T)}} - \overline{g}^{(T)}\|_{L_2(\rho_X)} + \|\overline{g}^{(T)} - g_\rho\|_{L_2(\rho_X)}.$$

That is, these two terms are bounded by Proposition A and Theorem B under Assumption **(A1)-(A3)**, resulting in Theorem 1, which exhibits comparable generalization error to Theorem B as long as the network width $M$ is sufficiently large.

Theorem 1 immediately leads to the fast convergence rate in Corollary 1 by setting $\eta_t = \eta = O(1)$ satisfying $4(6 + \lambda)\eta \leq 1$ and $\lambda = T^{-\beta/(2r\beta+1)}$ with the bounds on $\|g_\rho\|^2_{\mathcal{H}_\infty}$, $\|g_\rho\|^2_{L_2(\rho_X)}$, and the degree of freedom. Because $\beta$ in Assumption **(A4)** controls the complexity of the hypothesis space $\mathcal{H}_\infty$, it derives a bound on the degree of freedom, as shown in Caponnetto & De Vito (2007):

$$\mathrm{Tr}\left(\Sigma_\infty(\Sigma_\infty + \lambda I)^{-1}\right) = O(\lambda^{-1/\beta}).$$

In addition, the boundedness of $\|\Sigma_\infty\|_{\mathrm{op}} \leq O(1)$ gives

$$\|g_\rho\|_{\mathcal{H}_\infty} = \|\Sigma_\infty^{r-\frac{1}{2}}\Sigma_\infty^{-r}g_\rho\|_{L_2(\rho_X)} \leq O\left(\|\Sigma_\infty^{-r}g_\rho\|_{L_2(\rho_X)}\right),$$
$$\|g_\rho\|_{L_2(\rho_X)} \leq \|\Sigma_\infty^r\Sigma_\infty^{-r}g_\rho\|_{L_2(\rho_X)} \leq O\left(\|\Sigma_\infty^{-r}g_\rho\|_{L_2(\rho_X)}\right).$$

This finishes the proof of Corollary 1.

# B  PROOF OF PROPOSITION A

We first show the Proposition A that says the equivalence between averaged stochastic gradient descent for two-layer neural networks and that in an RKHS associated with $k_M$.

*Proof.* Proof of Proposition A

**Bound the growth of $\|g_{\Theta^{(t)}}\|_{L_\infty(\rho_X)}$.**  We first show that there exist increasing functions $d(t)$ and $M(t)$ depending only on $t$ uniformly over the choice of the history of examples $(x_t, y_t)_{t=1}^\infty$ used in Algorithms such that $\|g_{\Theta^{(s)}}\|_{L_\infty(\rho_X)} \leq d(t)$ for $\forall s \leq t$ when $M \geq M(t)$. We show this statement by the induction.

Without loss of generality, we assume that there is no bias term, $\|b_r^{(0)}\|_2 = 1$, and $\mathrm{supp}(\rho_X) \subset \{x \in \mathbb{R}^{d+1} \mid \|x\|_2 \leq 2\}$ by setting $x \leftarrow (x, \gamma)$ (where $\gamma \in (0, 1)$). Hence, we consider the update only for parameters $a$ and $B$. The above statement clearly holds for $t = 0$. Thus, we assume it holds for $t$. We recall the specific update rules of the stochastic gradient descent:

$$a_r^{(t+1)} - a_r^{(0)} = (1 - \eta\lambda)(a_r^{(t)} - a_r^{(0)}) - \frac{\eta}{\sqrt{M}}(g_{\Theta^{(t)}}(x_t) - y_t)\sigma(b_r^{(t)\top}x_t), \tag{9}$$

$$b_r^{(t+1)} - b_r^{(0)} = (1 - \eta\lambda)(b_r^{(t)} - b_r^{(0)}) - \frac{\eta}{\sqrt{M}}(g_{\Theta^{(t)}}(x_t) - y_t)a_r^{(t)}\sigma'(b_r^{(t)\top}x_t)x_t. \tag{10}$$

Here, let us consider $\forall M \geq M(t)$. Set $d_b^M(t) = \max_{s \leq t, 1 \leq r \leq M}\|b_r^{(s)}\|_2$. Then, by expanding equation (9), we get

$$|a_r^{(t+1)} - a_r^{(0)}| \leq |a_r^{(t)} - a_r^{(0)}| + \frac{\eta}{\sqrt{M}}(d(t) + 1)(1 + 2d_b^M(t))$$

$$\leq \frac{\eta}{\sqrt{M}}\sum_{s=0}^t (d(s) + 1)(1 + 2d_b^M(t))$$

$$\leq \frac{\eta(t + 1)}{\sqrt{M}}(d(t) + 1)(1 + 2d_b^M(t)), \tag{11}$$

where we used $\|\sigma(u)\| \leq 1 + u$ and $|y_t| \leq 1$. As for the term $|a_r^{(s)}|$ ($s \leq t + 1$), from the similar augment for $s$ and the monotonicity, we have for $s \leq t + 1$,

$$|a_r^{(s)}| \leq 1 + |a_r^{(s)} - a_r^{(0)}|$$

$$\leq 1 + \frac{\eta(t + 1)}{\sqrt{M}}(d(t) + 1)(1 + 2d_b^M(t))$$

$$\leq 1 + \eta(t + 1)(d(t) + 1)(1 + 2d_b^M(t)).$$

We next give a bound on $\|b_r^{(t+1)} - b_r^{(0)}\|_2$. By expanding equation (10), we get

$$\|b_r^{(t+1)} - b_r^{(0)}\|_2 \le \|b_r^{(t)} - b_r^{(0)}\|_2 + \frac{4\eta}{\sqrt{M}}|a_r^{(t)}|(d(t)+1)$$

$$\le \frac{4\eta}{\sqrt{M}}\sum_{s=0}^{t}|a_r^{(s)}|(d(s)+1)$$

$$\le \frac{4\eta(t+1)}{\sqrt{M}}(d(t)+1)\left(1 + \frac{\eta(t+1)}{\sqrt{M}}(d(t)+1)(1+2d_b^M(t))\right), \qquad (12)$$

where we used $\|\sigma'\|_\infty \le 2$ and $\|x_t\|_2 \le 2$. Here, we evaluate $d_b^M(t)$. From the similar augment for $s \le t$, the monotonicity, and $\|b_r^{(s)}\|_2 \le 1 + \|b_r^{(s)} - b_r^{(0)}\|_2$, we get

$$d_b^M(t) \le 1 + \frac{4\eta(t+1)}{\sqrt{M}}(d(t)+1)\left(1 + \frac{\eta(t+1)}{\sqrt{M}}(d(t)+1)(1+2d_b^M(t))\right).$$

Let $M'(t+1)$ be a positive integer depending on $t$ and $d(t)$ such that $\frac{t+1}{\sqrt{M}}(d(t)+1) \le \frac{1}{4}$. Let us reconsider $\forall M \ge M'(t+1)$. Then, since $\eta \le 1$, we have

$$d_b^M(t) \le \frac{5}{2}, \quad |a_r^{(s)}| \le \frac{5}{2} \quad (\forall s \le t+1).$$

From the derivation of (11) and (12) and since $\eta \le 1$, we have for $0 \le \forall s \le t+1$,

$$|a_r^{(s)} - a_r^{(0)}| \le \frac{d_1(t+1)}{\sqrt{M}}, \quad \|b_r^{(s)} - b_r^{(0)}\|_2 \le \frac{d_2(t+1)}{\sqrt{M}}, \qquad (13)$$

where $d_1(t+1)$ and $d_2(t+1)$ are set to

$$d_1(t+1) \overset{def}{=} 6(t+1)(d(t)+1), \quad d_2(t+1) \overset{def}{=} 10(t+1)(d(t)+1).$$

We next bound $|g_{\Theta^{(t+1)}}(x)|$ for $x \in \forall\mathrm{supp}(\rho_X)$ as follows. Since $g_{\Theta^{(0)}} \equiv 0$,

$$|g_{\Theta^{(t+1)}}(x)| = |g_{\Theta^{(t+1)}}(x) - g_{\Theta^{(0)}}(x)|$$

$$\le \frac{1}{\sqrt{M}}\sum_{r=1}^{M}\left\{\left|(a_r^{(t+1)} - a_r^{(0)})\sigma(b_r^{(0)\top}x)\right| + \left|a_r^{(t+1)}(\sigma(b_r^{(t+1)\top}x)) - \sigma(b_r^{(0)\top}x)\right|\right\}$$

$$\le \frac{1}{\sqrt{M}}\sum_{r=1}^{M}\left\{2\left|a_r^{(t+1)} - a_r^{(0)}\right| + 4\left|a_r^{(t+1)}\right|\left\|b_r^{(t+1)} - b_r^{(0)}\right\|\right\}$$

$$\le 2d_1(t+1) + 10d_2(t+1).$$

In summary, by setting $M(t+1) = \max\{M(t), 16(t+1)^2(d(t)+1)^2\}$ and $d(t+1) = 2d_1(t+1) + 10d_2(t+1)$, we get $\|g_{\Theta^{(t+1)}}\|_{L_\infty(\rho_X)} \le d(t+1)$ when $M \ge M(t+1)$. We note that from the above construction, $d(t), d_1(t), d_2(t)$ depend only on $t$ and inequalities (13) are always hold for $\forall t \in \mathbb{Z}_+$ when $M \ge M(t+1)$.

**Linear approximation of the model.** For a given $T \in \mathbb{Z}_+$, we consider $\forall M \ge M(T)$ and define the neighborhood of $\Theta^{(0)} = (a_r^{(0)}, b_r^{(0)})_{r=1}^{M}$:

$$B_T(\Theta^{(0)}) \overset{def}{=} \left\{(a_r, b_r)_{r=1}^{M} \in (\mathbb{R} \times \mathbb{R}^{d+1})^M \mid |a_r| \le \frac{5}{2}, |a_r - a_r^{(0)}| \le \frac{d_1(T)}{\sqrt{M}}, \|b_r - b_r^{(0)}\|_2 \le \frac{d_2(T)}{\sqrt{M}}\right\}.$$

From Taylor's formula $|\sigma(b_r^\top x) - \sigma(b_r^{(0)\top}x) - \sigma'(b_r^{(0)\top}x)(b_r - b_r^{(0)})^\top x| \le 2\|\sigma''\|_\infty\|b_r - b_r^{(0)}\|_2^2$ and the smoothness of $\sigma$, we get for $\Theta \in B_T(\Theta^{(0)})$ and $x \in \mathrm{supp}(\rho_X)$,

$$\left|a_r\sigma(b_r^\top x) - (a_r^{(0)}\sigma(b_r^{(0)\top}x) + (a_r - a_r^{(0)})\sigma(b_r^{(0)\top}x) + a_r^{(0)}\sigma'(b_r^{(0)\top}x)(b_r - b_r^{(0)})^\top x)\right|$$

$$\le 4|a_r - a_r^{(0)}|\|b_r - b_r^{(0)}\|_2 + 2C|a_r|\|b_r - b_r^{(0)}\|_2^2$$

$$\le \frac{2d_1(T)d_2(T)}{M} + \frac{5Cd_2^2(T)}{M}. \qquad (14)$$

We here define a linear model:

$$h_\Theta(x) \overset{def}{=} \frac{1}{\sqrt{M}} \sum_{r=1}^{M} \left( (a_r - a_r^{(0)})\sigma(b_r^{(0)\top}x) + a_r^{(0)}\sigma'(b_r^{(0)\top}x)(b_r - b_r^{(0)})^\top x \right).$$

By taking the sum of (14) over $r \in \{1, \ldots, M\}$ and by $g_{\Theta^{(0)}} \equiv 0$,

$$|g_\Theta(x) - h_\Theta(x)| \leq \frac{1}{\sqrt{M}} \sum_{r=1}^{M} \left( \frac{2d_1(T)d_2(T)}{M} + \frac{5Cd_2^2(T)}{M} \right)$$

$$\leq \frac{1}{\sqrt{M}} \left( 2d_1(T)d_2(T) + 5Cd_2^2(T) \right).$$

We denote $d_3(T) \overset{def}{=} 2d_1(T)d_2(T) + 5Cd_2^2(T)$. Since iterates $(\Theta^{(t)})_{t=0}^{T}$ obtained by Algorithm 1 are contained in $B_T(\Theta^{(0)})$, weighted averages $(\overline{\Theta}^{(t)})_{t=0}^{T}$ are also contained in $B_T(\Theta^{(0)})$. Thus, we get for $\forall t \in \{1, \ldots, T\}$,

$$|g_{\Theta^{(t)}}(x) - h_{\Theta^{(t)}}(x)| \leq \frac{d_3(T)}{\sqrt{M}}, \quad \left| g_{\overline{\Theta}^{(t)}}(x) - h_{\overline{\Theta}^{(t)}}(x) \right| \leq \frac{d_3(T)}{\sqrt{M}}. \tag{15}$$

**Recursion of $h_{\Theta^{(t)}}$ using the random feature approximation of NTK.** We here derive a recursion of $h_{\Theta^{(t)}}$ using $k_M$. From the updates (9) and (10), we have

$$h_{\Theta^{(t+1)}}(x) = \frac{1}{\sqrt{M}} \sum_{r=1}^{M} \left( (a_r^{(t+1)} - a_r^{(0)})\sigma(b_r^{(0)\top}x) + a_r^{(0)}\sigma'(b_r^{(0)\top}x)(b_r^{(t+1)} - b_r^{(0)})^\top x \right)$$

$$= (1 - \eta\lambda)h_{\Theta^{(t)}}(x) - \frac{\eta}{M} \sum_{r=1}^{M} (g_{\Theta^{(t)}}(x_t) - y_t)\sigma(b_r^{(t)\top}x_t)\sigma(b_r^{(0)\top}x)$$

$$- \frac{\eta}{M} \sum_{r=1}^{M} a_r^{(0)}\sigma'(b_r^{(0)\top}x)(g_{\Theta^{(t)}}(x_t) - y_t)a_r^{(t)}\sigma'(b_r^{(t)\top}x_t)x_t^\top x. \tag{16}$$

Note that for $t \in \{0, \ldots, T\}$,

$$|(g_{\Theta^{(t)}}(x_t) - y_t)\sigma(b_r^{(t)\top}x_t) - (h_{\Theta^{(t)}}(x_t) - y_t)\sigma(b_r^{(0)\top}x_t)|$$

$$\leq |(g_{\Theta^{(t)}}(x_t) - h_{\Theta^{(t)}}(x_t))\sigma(b_r^{(0)\top}x_t)| + |(g_{\Theta^{(t)}}(x_t) - y_t)(\sigma(b_r^{(t)\top}x_t) - \sigma(b_r^{(0)\top}x_t))|$$

$$\leq \frac{2d_3(T)}{\sqrt{M}} + 4(d(T) + 1)\|b_r^{(t)} - b_r^{(0)}\|_2$$

$$\leq \frac{2}{\sqrt{M}} \left( d_3(T) + 2(d(T) + 1)d_2(T) \right),$$

and

$$|(g_{\Theta^{(t)}}(x_t) - y_t)a_r^{(t)}\sigma'(b_r^{(t)\top}x_t) - (h_{\Theta^{(t)}}(x_t) - y_t)a_r^{(0)}\sigma'(b_r^{(0)\top}x_t)|$$

$$\leq |(g_{\Theta^{(t)}}(x_t) - h_{\Theta^{(t)}}(x_t))a_r^{(0)}\sigma'(b_r^{(0)\top}x_t)| + |(g_{\Theta^{(t)}}(x_t) - y_t)(a_r^{(t)}\sigma'(b_r^{(t)\top}x_t) - a_r^{(0)}\sigma'(b_r^{(0)\top}x_t))|$$

$$\leq \frac{2d_3(T)}{\sqrt{M}} + (d(T) + 1)|a_r^{(t)}\sigma'(b_r^{(t)\top}x_t) - a_r^{(0)}\sigma'(b_r^{(0)\top}x_t))|$$

$$\leq \frac{2d_3(T)}{\sqrt{M}} + (d(T) + 1)\left\{ |a_r^{(0)}(\sigma'(b_r^{(t)\top}x_t) - \sigma'(b_r^{(0)\top}x_t))| + |(a_r^{(t)} - a_r^{(0)})\sigma'(b_r^{(t)\top}x_t)| \right\}$$

$$\leq \frac{2d_3(T)}{\sqrt{M}} + 2(d(T) + 1)\left\{ C\|b_r^{(t)} - b_r^{(0)}\|_2 + |a_r^{(t)} - a_r^{(0)}| \right\}$$

$$\leq \frac{2d_3(T)}{\sqrt{M}} + \frac{2(d(T) + 1)}{\sqrt{M}}(d_1(T) + Cd_2(T))$$

$$= \frac{2}{\sqrt{M}}(d_3(T) + (d(T) + 1)(d_1(T) + Cd_2(T))).$$

Plugging these two inequalities into (16), we have $\forall t \in \{1, \ldots, T-1\}$,

$$
\begin{aligned}
h_{\Theta^{(t+1)}}(x) &\leq (1 - \eta\lambda)h_{\Theta^{(t)}}(x) - \frac{\eta}{M}\sum_{r=1}^{M}(h_{\Theta^{(t)}}(x_t) - y_t)\sigma(b_r^{(0)\top}x_t)\sigma(b_r^{(0)\top}x) \\
&\quad - \frac{\eta}{M}\sum_{r=1}^{M}\sigma'(b_r^{(0)\top}x)(h_{\Theta^{(t)}}(x_t) - y_t)\sigma'(b_r^{(0)\top}x_t)x_t^\top x \\
&\quad + \frac{2\eta}{\sqrt{M}}\left(2d_3(T) + (d(T)+1)\left(d_1(T) + (C+2)d_2(T)\right)\right) \\
&= (1 - \eta\lambda)h_{\Theta^{(t)}}(x) - \eta(h_{\Theta^{(t)}}(x_t) - y_t)\frac{1}{M}\sum_{r=1}^{M}\left(\sigma(b_r^{(0)\top}x_t)\sigma(b_r^{(0)\top}x) + \sigma'(b_r^{(0)\top}x)\sigma'(b_r^{(0)\top}x_t)x_t^\top x\right) \\
&\quad + \frac{2\eta}{\sqrt{M}}\left(2d_3(T) + (d(T)+1)\left(d_1(T) + (C+2)d_2(T)\right)\right) \\
&= (1 - \eta\lambda)h_{\Theta^{(t)}}(x) - \eta(h_{\Theta^{(t)}}(x_t) - y_t)k_M(x, x_t) + \frac{\eta}{\sqrt{M}}d_4(T),
\end{aligned}
$$

where $d_4(T) = 2d_3(T) + (d(T)+1)\left(d_1(T) + (C+2)d_2(T)\right)$. Clearly, the inverse inequality also holds:

$$
h_{\Theta^{(t+1)}}(x) \geq (1 - \eta\lambda)h_{\Theta^{(t)}}(x) - \eta(h_{\Theta^{(t)}}(x_t) - y_t)k_M(x, x_t) - \frac{\eta}{\sqrt{M}}d_4(T).
$$

Thus, we get

$$
\left|h_{\Theta^{(t+1)}}(x) - (1 - \eta\lambda)h_{\Theta^{(t)}}(x) + \eta(h_{\Theta^{(t)}}(x_t) - y_t)k_M(x, x_t)\right| \leq \frac{\eta}{\sqrt{M}}d_4(T). \qquad (17)
$$

**Equivalence between Algorithm 1 and 2.** We provide a bound between recursions of Algorithm 2 and (17). Noting that $h_{\Theta^{(0)}} \equiv g^{(0)} \equiv 0$, we have for $\forall t \in \{0, \ldots, T-1\}$,

$$
|h_{\Theta^{(t+1)}}(x) - g^{(t+1)}(x)| \leq (1 - \eta\lambda)|h_{\Theta^{(t)}}(x) - g^{(t)}(x)| + \eta|h_{\Theta^{(t)}}(x_t) - g^{(t)}(x_t)|k_M(x, x_t) + \frac{\eta}{\sqrt{M}}d_4(T).
$$

Noting $\|k_M\|_{L_\infty(\rho_X)} \leq 12$ and taking a supremum over $x, x_t \in \mathrm{supp}(\rho_X)$ in both sides, we have

$$
\begin{aligned}
\|h_{\Theta^{(t+1)}} - g^{(t+1)}\|_{L_\infty(\rho_X)} &\leq (1 - \eta\lambda)\|h_{\Theta^{(t)}} - g^{(t)}\|_{L_\infty(\rho_X)} + \eta\|h_{\Theta^{(t)}} - g^{(t)}\|_{L_\infty(\rho_X)}\|k_M\|_{L_\infty(\rho_X)} + \frac{\eta}{\sqrt{M}}d_4(T) \\
&\leq (1 - \eta\lambda + 12\eta)\|h_{\Theta^{(t)}} - g^{(t)}\|_{L_\infty(\rho_X)} + \frac{\eta}{\sqrt{M}}d_4(T) \\
&\leq \sum_{s=0}^{t}(1 + 12\eta)^{t-s}\frac{\eta}{\sqrt{M}}d_4(T) \\
&\leq \frac{T}{\sqrt{M}}(1 + 12\eta)^T d_4(T).
\end{aligned}
$$

Since $h_\Theta$ is a linear model, we have $h_{\overline{\Theta}^{(T)}} = \sum_{t=0}^{T}\alpha_t h_{\Theta^{(t)}}$ and

$$
\|h_{\overline{\Theta}^{(T)}} - \overline{g}^{(T)}\|_{L_\infty(\rho_X)} \leq \sum_{t=0}^{T}\alpha_t\|h_{\Theta^{(t)}} - g^{(t)}\|_{L_\infty(\rho_X)} \leq \frac{T}{\sqrt{M}}(1 + 12\eta)^T d_4(T).
$$

Combining this inequality with (15), we finally have

$$
\begin{aligned}
\|g_{\overline{\Theta}^{(T)}} - \overline{g}^{(T)}\|_{L_\infty(\rho_X)} &\leq \|g_{\overline{\Theta}^{(T)}} - h_{\overline{\Theta}^{(T)}}\|_{L_\infty(\rho_X)} + \|h_{\overline{\Theta}^{(T)}} - \overline{g}^{(T)}\|_{L_\infty(\rho_X)} \\
&\leq \frac{1}{\sqrt{M}}(d_3(T) + 13^T T d_4(T)).
\end{aligned}
$$

Because $(d_3(T) + 13^T T d_4(T))$ depends only on $T$ and $C$ from the construction, $\|g_{\Theta^{(T)}} - \overline{g}^{(T)}\|_{L_\infty(\rho_X)} \to 0$ as $M \to \infty$. This finishes the proof of Proposition A. $\qquad\square$

## C    PROOF OF THEOREM A

In this section, we give the proof of the convergence theory for the reference ASGD (Algorithm 2). We introduce an auxiliary result for proving Theorem A.

**Lemma A.** *Suppose Assumption (**A1**), (**A2**), and (**A3**) hold. Set $\xi \stackrel{def}{=} Y K_{M,X} - (K_{M,X} \otimes_{\mathcal{H}_M} K_{M,X} + \lambda I) g_{M,\lambda}$. Then, for $\forall \lambda > 0$ and $\forall \delta \in (0,1)$ there exists $M_0 > 0$ such that for $\forall M \geq M_0$ the following holds with high probability at least $1 - \delta$:*

$$\mathbb{E}_{(X,Y)\sim\rho}[\xi \otimes_{\mathcal{H}_M} \xi] \preccurlyeq 2(1 + \|g_\rho\|^2_{L_2(\rho_X)} + 24\|\Sigma_\infty^{-r} g_\rho\|^2_{L_2(\rho_X)})\Sigma_M.$$

*Proof.* Since $\xi = (Y - g_{M,\lambda}(X))K_{M,X} - \lambda g_{M,\lambda}$, we get

$$\begin{aligned}
\mathbb{E}[\xi \otimes_{\mathcal{H}_M} \xi] = {} & \mathbb{E}[(Y - g_{M,\lambda}(X))^2 K_{M,X} \otimes_{\mathcal{H}_M} K_{M,X}] \\
& - \lambda \mathbb{E}[(Y - g_{M,\lambda}(X))K_{M,X}] \otimes_{\mathcal{H}_M} g_{M,\lambda} \\
& - \lambda g_{M,\lambda} \otimes_{\mathcal{H}_M} \mathbb{E}[(Y - g_{M,\lambda}(X))K_{M,X}] \\
& + \lambda^2 g_{M,\lambda} \otimes_{\mathcal{H}_M} g_{M,\lambda}.
\end{aligned}$$

We evaluate an expectation in the second and third terms in the right hand side of the above equation as follows:

$$\begin{aligned}
\mathbb{E}[(Y - g_{M,\lambda}(X))K_{M,X}] &= \mathbb{E}[Y K_{M,X} - (K_{M,X} \otimes_{\mathcal{H}_M} K_{M,X}) g_{M,\lambda}] \\
&= \mathbb{E}[Y K_{M,X}] - \Sigma_M g_{M,\lambda} \\
&= \mathbb{E}[Y K_{M,X}] - \Sigma_M (\Sigma_M + \lambda I)^{-1} \mathbb{E}[Y K_{M,X}] \\
&= \mathbb{E}[Y K_{M,X}] - (\Sigma_M + \lambda I - \lambda I)(\Sigma_M + \lambda I)^{-1} \mathbb{E}[Y K_{M,X}] \\
&= \lambda (\Sigma_M + \lambda I)^{-1} \mathbb{E}[Y K_{M,X}] \\
&= \lambda g_{M,\lambda}.
\end{aligned}$$

Hence, we get

$$\mathbb{E}[\xi \otimes_{\mathcal{H}_M} \xi] \preccurlyeq \mathbb{E}[(Y - g_{M,\lambda}(X))^2 K_{M,X} \otimes_{\mathcal{H}_M} K_{M,X}].$$

For $h \in \mathcal{H}_M$,

$$\begin{aligned}
\big\langle \mathbb{E}[(Y &- g_{M,\lambda}(X))^2 K_{M,X} \otimes_{\mathcal{H}_M} K_{M,X}]h, h \big\rangle_{\mathcal{H}_M} \\
&= \mathbb{E}[(Y - g_{M,\lambda}(X))^2 \langle (K_{M,X} \otimes_{\mathcal{H}_M} K_{M,X})h, h \rangle_{\mathcal{H}_M}] \\
&\leq \|Y - g_{M,\lambda}(X)\|^2_{L_\infty(\rho_X)} \mathbb{E}[\langle (K_{M,X} \otimes_{\mathcal{H}_M} K_{M,X})h, h \rangle_{\mathcal{H}_M}] \\
&\leq 2(1 + \|g_{M,\lambda}\|^2_{L_\infty(\rho_X)}) \mathbb{E}[\langle (K_{M,X} \otimes_{\mathcal{H}_M} K_{M,X})h, h \rangle_{\mathcal{H}_M}], \qquad (18)
\end{aligned}$$

where we used Assumption **(A2)** for the last inequality.

Finally, we provide an upper-bound on $\|g_{M,\lambda}\|_{L_\infty(\rho_X)}$. Since $S^{-1} - T^{-1} = -S^{-1}(S - T)T^{-1}$ for arbitrary operators $S$ and $T$, we get

$$\begin{aligned}
\| \big( (\Sigma_\infty &+ \lambda I)^{-1} - (\Sigma_M + \lambda I)^{-1} \big) g_\rho \|_{L_2(\rho_X)} \\
&= \|(\Sigma_\infty + \lambda I)^{-1}(\Sigma_\infty - \Sigma_M)(\Sigma_M + \lambda I)^{-1} g_\rho\|_{L_2(\rho_X)} \\
&= \|(\Sigma_\infty + \lambda I)^{-1}\|_{\mathrm{op}} \|\Sigma_\infty - \Sigma_M\|_{\mathrm{op}} \|(\Sigma_M + \lambda I)^{-1}\|_{\mathrm{op}} \|g_\rho\|_{L_2(\rho_X)} \\
&\leq \frac{1}{\lambda^2} \|\Sigma_\infty - \Sigma_M\|_{\mathrm{op}} \|g_\rho\|_{L_2(\rho_X)}. \qquad (19)
\end{aligned}$$

We denote $F_\infty = (\Sigma_\infty + \lambda I)^{-1} g_\rho$ and $F_M = (\Sigma_M + \lambda I)^{-1} g_\rho$. Noting $g_{\infty,\lambda} = \Sigma_\infty F_\infty$ and $g_{M,\lambda} = \Sigma_M F_M$, we get for $\forall x \in \mathrm{supp}(\rho_X)$,

$$|g_{\infty,\lambda}(x) - g_{M,\lambda}(x)| = |\Sigma_\infty F_\infty(x) - \Sigma_M F_M(x)|$$

$$= \left| \int_{\mathcal{X}} K_{\infty,x}(X) F_\infty(X) \mathrm{d}\rho_X - \int_{\mathcal{X}} K_{M,x}(X) F_M(X) \mathrm{d}\rho_X \right|$$

$$= \left| \int_{\mathcal{X}} (K_{\infty,x} - K_{M,x})(X) F_\infty(X) \mathrm{d}\rho_X - \int_{\mathcal{X}} K_{M,x}(X)(F_M(X) - F_\infty(X)) \mathrm{d}\rho_X \right|$$

$$\leq \|K_{\infty,x} - K_{M,x}\|_{L_2(\rho_X)} \|F_\infty\|_{L_2(\rho_X)} + \|K_{M,x}\|_{L_2(\rho_X)} \|F_M - F_\infty\|_{L_2(\rho_X)}$$

$$\leq \frac{1}{\lambda} \|k_\infty - k_M\|_{L_\infty(\rho_X)^2} \|g_\rho\|_{L_2(\rho_X)} + \frac{12}{\lambda^2} \|\Sigma_\infty - \Sigma_M\|_{\mathrm{op}} \|g_\rho\|_{L_2(\rho_X)},$$

where we used $k_M(x, x') \leq 12$ for $\forall (x, x') \in \text{supp}(\rho_X) \times \text{supp}(\rho_X)$ and inequality (19).

Moreover, we get

$$
\begin{aligned}
|g_{\infty,\lambda}(x)| = | \langle g_{\infty,\lambda}, K_x \rangle_{\mathcal{H}_\infty} | \\
\leq \|K_x\|_{\mathcal{H}_\infty} \|g_{\infty,\lambda}\|_{\mathcal{H}_\infty} \\
\leq 2\sqrt{3} \|\Sigma_\infty (\Sigma_\infty + \lambda I)^{-1} g_\rho\|_{\mathcal{H}_\infty} \\
\leq 2\sqrt{3} \|\Sigma_\infty^{1+r} (\Sigma_\infty + \lambda I)^{-1} \Sigma_\infty^{-r} g_\rho\|_{\mathcal{H}_\infty} \\
\leq 2\sqrt{3} \|\Sigma_\infty^{\frac{1}{2}+r} (\Sigma_\infty + \lambda I)^{-1} \Sigma_\infty^{-r} g_\rho\|_{L_2(\rho_X)} \\
\leq 2\sqrt{3} \|\Sigma_\infty^{\frac{1}{2}+r} (\Sigma_\infty + \lambda I)^{-1}\|_{\text{op}} \|\Sigma_\infty^{-r} g_\rho\|_{L_2(\rho_X)} \\
\leq 2\sqrt{3} \|\Sigma_\infty^{-r} g_\rho\|_{L_2(\rho_X)}.
\end{aligned}
\tag{20}
$$

where we used Assumption (**A3**) and the isometric map $\Sigma_\infty^{1/2} : L_2(\rho_X) \to \mathcal{H}_\infty$.

Hence, we get

$$
\|g_{M,\lambda}\|_{L_\infty(\rho_X)} \leq \left( \frac{1}{\lambda} \|k_\infty - k_M\|_{L_\infty(\rho_X)^2} + \frac{12}{\lambda^2} \|\Sigma_\infty - \Sigma_M\|_{\text{op}} \right) \|g_\rho\|_{L_2(\rho_X)} + 2\sqrt{3} \|\Sigma_\infty^{-r} g_\rho\|_{L_2(\rho_X)}.
$$

By the uniform law of large numbers (Theorem 3.1 in Mohri et al. (2012)) and the Bernstein's inequality (Proposition 3 in Rudi & Rosasco (2017)) to random operators, $\|k_\infty - k_M\|_{L_\infty(\rho_X \times \rho_X)}$ and $\|\Sigma_\infty - \Sigma_M\|_{\text{op}}$ converge to zero as $M \to \infty$ in probability. That is, for given $\lambda > 0$ and $\delta \in (0, 1)$, there exists $M_0$ such that for any $M \geq M_0$ the following holds with high probability at least $1 - \delta$:

$$
\|g_{M,\lambda}\|_{L_\infty(\rho_X)} \leq \frac{1}{2} \|g_\rho\|_{L_2(\rho_X)} + 2\sqrt{3} \|\Sigma_\infty^{-r} g_\rho\|_{L_2(\rho_X)}.
$$

Combining with (18), we get

$$
\begin{aligned}
\langle \mathbb{E}[(Y - g_{M,\lambda}(X))^2 K_{M,X} \otimes_{\mathcal{H}_M} K_{M,X}] h, h \rangle_{\mathcal{H}_M} \\
\leq 2(1 + \|g_\rho\|_{L_2(\rho_X)}^2 + 24 \|\Sigma_\infty^{-r} g_\rho\|_{L_2(\rho_X)}^2) \mathbb{E}[\langle (K_{M,X} \otimes_{\mathcal{H}_M} K_{M,X}) h, h \rangle_{\mathcal{H}_M}].
\end{aligned}
$$

$\square$

*Proof of Theorem A.* Since, stochastic gradient in $\mathcal{H}_M$ is described as

$$
G^{(t)} = \partial_z \ell(g^{(t)}(x_t), y_t) k_M(x_t, \cdot) = \left( \left\langle K_{M,x_t}, g^{(t)} \right\rangle_{\mathcal{H}_M} - y_t \right) K_{M,x_t},
$$

the update rule of Algorithm 2 is

$$
\begin{aligned}
g^{(t+1)} = (1 - \eta\lambda) g^{(t)} - \eta \left( \left\langle K_{M,x_t}, g^{(t)} \right\rangle_{\mathcal{H}_M} - y_t \right) K_{M,x_t} \\
= (I - \eta K_{M,x_t} \otimes_{\mathcal{H}_M} K_{M,x_t} - \eta\lambda I) g^{(t)} + \eta y_t K_{M,x_t}.
\end{aligned}
$$

Hence, we get

$$
\begin{aligned}
g^{(t+1)} - g_{M,\lambda} = \underbrace{(I - \eta K_{M,x_t} \otimes_{\mathcal{H}_M} K_{M,x_t} - \eta\lambda I)}_{=\alpha_t} \underbrace{(g^{(t)} - g_{M,\lambda})}_{=A_t} \\
\underbrace{- \eta (K_{M,x_t} \otimes_{\mathcal{H}_M} K_{M,x_t} + \lambda I) g_{M,\lambda} + \eta y_t K_{M,x_t}}_{=\beta_t}.
\end{aligned}
\tag{21}
$$

This leads to the following stochastic recursion: $t \in \{0, \dots, T-1\}$,

$$
A_{t+1} = \alpha_t A_t + \beta_t = \prod_{s=0}^{t} \alpha_s A_0 + \sum_{s=0}^{t} \prod_{l=s+1}^{t} \alpha_s \beta_s.
$$

By taking the average, we get

$$\overline{A}_T = \frac{1}{T+1} \sum_{t=0}^{T} A_t$$

$$= \underbrace{\frac{1}{T+1} \sum_{t=0}^{T} \prod_{s=0}^{t} \alpha_s A_0}_{\text{Bias term}} + \underbrace{\frac{1}{T+1} \sum_{t=0}^{T} \sum_{s=0}^{t} \prod_{l=s+1}^{t} \alpha_s \beta_s}_{\text{Noise term}} . \tag{22}$$

Thus, the average $\overline{A}_T$ is composed of bias and noise terms. We next bound these two terms, separately.

**Bound the bias term.** Note that the bias term exactly corresponds to the recursion (21) with $\beta_t = 0$. Hence, we consider the case of $\beta_t = 0$ and consider the following stochastic recursion in $\mathcal{H}_M$: $A_0 = -g_{M,\lambda}$,

$$A_{t+1} = (I - \eta H_t - \eta \lambda I) A_t,$$

where we define $H_t = K_{M,x_t} \otimes_{\mathcal{H}_M} K_{M,x_t}$. In addition, we consider the deterministic recursion of this recursion: $A_0' = A_0$,

$$A_{t+1}' = (I - \eta \Sigma_M - \eta \lambda I) A_t'.$$

We set

$$\overline{A}_T = \frac{1}{T+1} \sum_{t=0}^{T} A_t, \quad \overline{A'}_T = \frac{1}{T+1} \sum_{t=0}^{T} A_t'.$$

Then, the bias term we want to evaluate is decomposed as follows: by Minkowski's inequality,

$$\left( \mathbb{E}[\|\overline{A}_T\|_{L_2(\rho_X)}^2] \right)^{\frac{1}{2}} \leq \|\overline{A'}_T\|_{L_2(\rho_X)} + \left( \mathbb{E}[\|\overline{A}_T - \overline{A'}_T\|_{L_2(\rho_X)}^2] \right)^{\frac{1}{2}}. \tag{23}$$

We here bound the first term in the right hand side of (23). Note that from $4(6+\lambda)\eta \leq 1$ and $\|k_M\|_{L_\infty(\rho_X)^2} \leq 12$, we see [1] $\eta(\Sigma_M + \lambda I) \preccurlyeq \eta(12+\lambda)I \prec \frac{1}{2}I$. Since $A_t' = (I - \eta \Sigma_M - \eta \lambda I)^t g_{M,\lambda}$, its average is

$$\overline{A'}_T = \frac{1}{T+1} \sum_{t=0}^{T} A_t' = \frac{1}{\eta(T+1)} (\Sigma_M + \lambda I)^{-1} (I - (I - \eta \Sigma_M - \eta \lambda)^{T+1}) g_{M,\lambda}.$$

Therefore,

$$\|\overline{A'}_T\|_{L_2(\rho_X)} = \frac{1}{\eta(T+1)} \|(\Sigma_M + \lambda I)^{-1} (I - (I - \eta \Sigma_M - \eta \lambda)^{T+1}) g_{M,\lambda}\|_{L_2(\rho_X)}$$

$$\leq \frac{1}{\eta(T+1)} \|(\Sigma_M + \lambda I)^{-1} g_{M,\lambda}\|_{L_2(\rho_X)}. \tag{24}$$

We bound the second term in (23), which measures the gap between $\overline{A}_T$ and $\overline{A'}_T$. To do so, we consider the following recursion:

$$A_{t+1} - A_{t+1}' = A_t - A_t' - \eta(H_t + \lambda I)(A_t - A_t') + \eta(\Sigma_M - H_t)A_t'.$$

Hence, we have

$$\|A_{t+1} - A_{t+1}'\|_{\mathcal{H}_M}^2 = \|A_t - A_t'\|_{\mathcal{H}_M}^2$$
$$- \eta \langle A_t - A_t', (H_t + \lambda I)(A_t - A_t') - (\Sigma_M - H_t)A_t' \rangle_{\mathcal{H}_M}$$
$$- \eta \langle (H_t + \lambda I)(A_t - A_t') - (\Sigma_M - H_t)A_t', A_t - A_t' \rangle_{\mathcal{H}_M}$$
$$+ \eta^2 \|(H_t + \lambda I)(A_t - A_t') - (\Sigma_M - H_t)A_t'\|_{\mathcal{H}_M}^2.$$

---

[1] In general, for any operator $F : L_2(\rho_X) \to L_2(\rho_X)$ that commutes with $\Sigma_M$ and has a common eigen-bases with $\Sigma_M$, it follows that $F(\mathcal{H}_M) \subset \mathcal{H}_M$ and inequality $F \succcurlyeq 0$ in $L_2(\rho_X)$ is equivalent with $F|_{\mathcal{H}_M} \succcurlyeq 0$. Hence, we do not specify a Hilbert space we consider in such a case for the simplicity.

Let $(\mathcal{F}_t)_{t=0}^{T-1}$ be a filtration. We take a conditional expectation given $\mathcal{F}_t$:

$$\mathbb{E}[\|A_{t+1} - A'_{t+1}\|^2_{\mathcal{H}_M} \mid \mathcal{F}_t] \leq \|A_t - A'_t\|^2_{\mathcal{H}_M} - 2\eta \langle (\Sigma_M + \lambda I)(A_t - A'_t), A_t - A'_t \rangle_{\mathcal{H}_M}$$

$$+ 2\eta^2 \mathbb{E}[\|(H_t + \lambda I)(A_t - A'_t)\|^2_{\mathcal{H}_M} \mid \mathcal{F}_t] \tag{25}$$

$$+ 2\eta^2 \mathbb{E}[\|(\Sigma_M - H_t)A'_t\|^2_{\mathcal{H}_M} \mid \mathcal{F}_t], \tag{26}$$

where we used $\|g + h\|^2_{\mathcal{H}_M} \leq 2(\|g\|^2_{\mathcal{H}_M} + \|h\|^2_{\mathcal{H}_M})$.

For $g \in \mathcal{H}_M$, we have

$$\langle \mathbb{E}[(K_{M,x_t} \otimes_{\mathcal{H}_M} K_{M,x_t})^2]g, g \rangle_{\mathcal{H}_M} = \mathbb{E}\left[\langle (K_{M,x_t} \otimes_{\mathcal{H}_M} K_{M,x_t})^2 g, g \rangle_{\mathcal{H}_M}\right]$$

$$= \mathbb{E}\left[\langle \langle K_{M,x_t}, g \rangle_{\mathcal{H}_M} (K_{M,x_t} \otimes_{\mathcal{H}_M} K_{M,x_t})K_{M,x_t}, g \rangle_{\mathcal{H}_M}\right]$$

$$= \mathbb{E}\left[\langle \langle K_{M,x_t}, g \rangle_{\mathcal{H}_M} k_M(x_t, x_t)K_{M,x_t}, g \rangle_{\mathcal{H}_M}\right]$$

$$= \mathbb{E}\left[\langle K_{M,x_t}, g \rangle^2_{\mathcal{H}_M} k_M(x_t, x_t)\right]$$

$$\leq 12\mathbb{E}\left[\langle K_{M,x_t}, g \rangle^2_{\mathcal{H}_M}\right]$$

$$= 12 \langle \mathbb{E}[K_{M,x_t} \otimes_{\mathcal{H}_M} K_{M,x_t}]g, g \rangle_{\mathcal{H}_M}. \tag{27}$$

where we used $k_M(x_t, x_t) \leq 12$ which is confirmed from the definition of $k_M$ and Assumption (A2). This means that $\mathbb{E}[H_t^2] = \mathbb{E}[(K_{M,x_t} \otimes_{\mathcal{H}_M} K_{M,x_t})^2] \preccurlyeq 12\Sigma_M$ on $\mathcal{H}_M \times \mathcal{H}_M$. Hence, we get a bound on (25) as follows:

$$\mathbb{E}[\|(H_t + \lambda I)(A_t - A'_t)\|^2_{\mathcal{H}_M} \mid \mathcal{F}_t] = \mathbb{E}[\langle (H_t + \lambda I)^2(A_t - A'_t), A_t - A'_t \rangle_{\mathcal{H}_M} \mid \mathcal{F}_t]$$

$$= \langle (\lambda^2 I + 2\lambda \Sigma_M + \mathbb{E}[(K_{M,x_t} \otimes_{\mathcal{H}_M} K_{M,x_t})^2]) (A_t - A'_t), A_t - A'_t \rangle_{\mathcal{H}_M}$$

$$\leq \langle (\lambda^2 I + 2(6 + \lambda)\Sigma_M) (A_t - A'_t), A_t - A'_t \rangle_{\mathcal{H}_M}.$$

Next, we bound a term (26):

$$\mathbb{E}[\|(\Sigma_M - H_t)A'_t\|^2_{\mathcal{H}_M} \mid \mathcal{F}_t] = \mathbb{E}[\langle (\Sigma_M - H_t)^2 A'_t, A'_t \rangle_{\mathcal{H}_M} \mid \mathcal{F}_t]$$

$$= \mathbb{E}[\langle (\Sigma_M^2 - \Sigma_M H_t - H_t \Sigma_M + H_t^2)A'_t, A'_t \rangle_{\mathcal{H}_M} \mid \mathcal{F}_t]$$

$$= \langle (\mathbb{E}[H_t^2] - \Sigma_M^2)A'_t, A'_t \rangle_{\mathcal{H}_M}$$

$$\leq \langle \mathbb{E}[H_t^2]A'_t, A'_t \rangle_{\mathcal{H}_M}$$

$$\leq 12 \langle \Sigma_M A'_t, A'_t \rangle_{\mathcal{H}_M}.$$

Combining these inequalities, we get

$$\mathbb{E}[\|A_{t+1} - A'_{t+1}\|^2_{\mathcal{H}_M} \mid \mathcal{F}_t] \leq \|A_t - A'_t\|^2_{\mathcal{H}_M} - 2\eta \langle (\Sigma_M + \lambda I)(A_t - A'_t), A_t - A'_t \rangle_{\mathcal{H}_M}$$

$$+ 2\eta^2 \langle (\lambda^2 I + 2(6 + \lambda)\Sigma_M) (A_t - A'_t), A_t - A'_t \rangle_{\mathcal{H}_M}$$

$$+ 24\eta^2 \langle \Sigma_M A'_t, A'_t \rangle_{\mathcal{H}_M}$$

$$= (1 - 2\lambda\eta + 2\lambda^2\eta^2)\|A_t - A'_t\|^2_{\mathcal{H}_M} - 2\eta \langle \Sigma_M(A_t - A'_t), A_t - A'_t \rangle_{\mathcal{H}_M}$$

$$+ 4\eta^2(6 + \lambda) \langle \Sigma_M(A_t - A'_t), A_t - A'_t \rangle_{\mathcal{H}_M}$$

$$+ 24\eta^2 \langle \Sigma_M A'_t, A'_t \rangle_{\mathcal{H}_M}$$

$$\leq \|A_t - A'_t\|^2_{\mathcal{H}_M} - \eta \langle \Sigma_M(A_t - A'_t), A_t - A'_t \rangle_{\mathcal{H}_M}$$

$$+ 24\eta^2 \langle \Sigma_M A'_t, A'_t \rangle_{\mathcal{H}_M}, \tag{28}$$

where for the last inequality we used $4\eta(6 + \lambda) \leq 1$.

By taking the expectation and the average of (28) over $t \in \{0, \ldots, T - 1\}$, we get

$$\frac{1}{T+1} \sum_{t=0}^{T} \langle \Sigma_M(A_t - A'_t), A_t - A'_t \rangle_{\mathcal{H}_M} \leq \frac{24\eta}{T+1} \sum_{t=0}^{T} \langle \Sigma_M A'_t, A'_t \rangle_{\mathcal{H}_M}.$$

Since $\Sigma_M^{1/2} : L_2(\rho_X) \to \mathcal{H}_M$ is isometric, we see $\|\Sigma_M^{1/2}(A_t - A_t')\|_{\mathcal{H}_M} = \|A_t - A_t'\|_{L_2(\rho_X)}$. Thus, the second term in (23) can be bounded as follows:

$$
\begin{aligned}
\mathbb{E}[\|\overline{A}_T - \overline{A'}_T\|^2_{L_2(\rho_X)}] = \mathbb{E}[\|\Sigma_M^{1/2}(\overline{A}_T - \overline{A'}_T)\|^2_{\mathcal{H}_M}] \\
\le \frac{1}{T+1} \sum_{t=0}^{T} \mathbb{E}[\|\Sigma_M^{1/2}(A_t - A_t')\|^2_{\mathcal{H}_M}] \\
\le \frac{24\eta}{T+1} \sum_{t=0}^{T} \|\Sigma_M^{1/2} A_t'\|^2_{\mathcal{H}_M} \\
= \frac{24\eta}{T+1} \sum_{t=0}^{T} \|\Sigma_M^{1/2}(I - \eta\Sigma_M - \eta\lambda I)^t g_{M,\lambda}\|^2_{\mathcal{H}_M} \\
= \frac{24\eta}{T+1} \left\langle \sum_{t=0}^{T}(I - \eta\Sigma_M - \eta\lambda I)^{2t}\Sigma_M^{1/2} g_{M,\lambda}, \Sigma_M^{1/2} g_{M,\lambda} \right\rangle_{\mathcal{H}_M} \\
\le \frac{24}{T+1} \left\langle (\Sigma_M + \lambda I)^{-1}\Sigma_M^{1/2} g_{M,\lambda}, \Sigma_M^{1/2} g_{M,\lambda} \right\rangle_{\mathcal{H}_M} \\
= \frac{24}{T+1} \|(\Sigma_M + \lambda I)^{-1/2} g_{M,\lambda}\|^2_{L_2(\rho_X)} \quad (29)
\end{aligned}
$$

where we used the convexity for the first inequality and we used the following inequality for the last inequality: since $\|k_M\|_{L_\infty(\rho_X)^2} \le 12$ and $\eta(\Sigma_M + \lambda I) \preccurlyeq \eta(12 + \lambda)I \preccurlyeq \frac{1}{2}I$,

$$
\sum_{t=0}^{T}(I - \eta\Sigma_M - \eta\lambda I)^{2t} \preccurlyeq \frac{1}{\eta}(\Sigma_M + \lambda I)^{-1}.
$$

By plugging (24) and (29) into (23), we get the bound on the bias term:

$$
\begin{aligned}
\mathbb{E}[\|\overline{A}_T\|^2_{L_2(\rho_X)}] \le 2\|\overline{A'}_T\|^2_{L_2(\rho_X)} + 2\mathbb{E}[\|\overline{A}_T - \overline{A'}_T\|^2_{L_2(\rho_X)}] \\
\le \frac{2}{\eta^2(T+1)^2}\|(\Sigma_M + \lambda I)^{-1} g_{M,\lambda}\|^2_{L_2(\rho_X)} + \frac{24^2}{T+1}\|(\Sigma_M + \lambda I)^{-1/2} g_{M,\lambda}\|^2_{L_2(\rho_X)}.
\end{aligned}
$$
(30)

**Bound the noise term.** Note that the noise term in (22) exactly corresponds to the recursion (21) with $A_0 = 0$. Hence, it is enough to consider the case of $A_0 = 0$ to evaluate the noise term. In this case, the average $\overline{A}_T$ can be rewritten as follows:

$$
\overline{A}_T = \frac{1}{T+1}\sum_{t=0}^{T}\sum_{s=0}^{t}\prod_{l=s+1}^{t}\alpha_l\beta_s = \frac{\eta}{T+1}\sum_{s=0}^{T}\underbrace{\sum_{t=s}^{T}\prod_{l=s+1}^{t}\alpha_l\frac{\beta_s}{\eta}}_{=Z_s}.
$$

We here evaluate the noise term. We set $z_s = (x_s, y_s)$. Note that since $\mathbb{E}_{z_s}[\beta_s] = 0$, we have for $s < s'$,

$$
\begin{aligned}
\mathbb{E}_{(z_s,\ldots,z_T)}\left[\langle Z_s, Z_{s'}\rangle_{L_2(\rho_X)}\right] = \int_{\mathcal{X}} \mathbb{E}_{(z_s,\ldots,z_T)}\left[\left(\sum_{t=s}^{T}\prod_{l=s+1}^{t}\alpha_l\frac{\beta_s}{\eta}\right)\left(\sum_{t=s'}^{T}\prod_{l=s'+1}^{t}\alpha_l\frac{\beta_{s'}}{\eta}\right)\right]d\rho_X \\
= \int_{\mathcal{X}} \mathbb{E}_{z_s}[\beta_s]\mathbb{E}_{(z_{s+1},\ldots,z_T)}\left[\beta_{s'}\left(\sum_{t=s}^{T}\prod_{l=s+1}^{t}\frac{\alpha_l}{\eta}\right)\left(\sum_{t=s'}^{T}\prod_{l=s'+1}^{t}\frac{\alpha_l}{\eta}\right)\right]d\rho_X \\
= 0.
\end{aligned}
$$

Therefore, we have

$$
\mathbb{E}[\|\overline{A}_T\|^2_{L_2(\rho_X)}] = \frac{\eta^2}{(T+1)^2}\mathbb{E}\left[\left\|\sum_{s=0}^{T} Z_s\right\|^2_{L_2(\rho_X)}\right]
$$

$$
= \frac{\eta^2}{(T+1)^2}\mathbb{E}\left[\sum_{s,s'=0}^{T} \langle Z_s, Z_{s'}\rangle_{L_2(\rho_X)}\right]
$$

$$
= \frac{\eta^2}{(T+1)^2}\sum_{s=0}^{T}\mathbb{E}\left[\langle Z_s, Z_s\rangle_{L_2(\rho_X)}\right]
$$

$$
= \frac{\eta^2}{(T+1)^2}\sum_{s=0}^{T}\mathbb{E}\left[\|\Sigma_M^{1/2}Z_s\|^2_{\mathcal{H}_M}\right]. \tag{31}
$$

Here, we apply Lemma 21 in Pillaud-Vivien et al. (2018a) with $A = \Sigma_M$, $H = \Sigma_M + \lambda I$, $C = 2(1 + \|g_\rho\|^2_{L_2(\rho_X)} + 24\|\Sigma_\infty^{-r}g_\rho\|^2_{L_2(\rho_X)})\Sigma_M$. One of required conditions in this lemma is verified by Lemma A. We verify the other required condition described below:

$$
\mathbb{E}\left[(K_{M,X}\otimes_{\mathcal{H}} K_{M,X} + \lambda I)CH^{-1}(K_{M,X}\otimes_{\mathcal{H}} K_{M,X} + \lambda I)\right] \preccurlyeq \frac{1}{\eta}C. \tag{32}
$$

Indeed, we have

$$
\mathbb{E}\left[(K_{M,X}\otimes_{\mathcal{H}} K_{M,X} + \lambda I)CH^{-1}(K_{M,X}\otimes_{\mathcal{H}} K_{M,X} + \lambda I)\right]
$$

$$
= \mathbb{E}\left[K_{M,X}\otimes_{\mathcal{H}} K_{M,X}CH^{-1}K_{M,X}\otimes_{\mathcal{H}} K_{M,X}\right] + 2\lambda\Sigma_M CH^{-1} + \lambda^2 CH^{-1}
$$

$$
\preccurlyeq \mathbb{E}\left[K_{M,X}\otimes_{\mathcal{H}} K_{M,X}CH^{-1}K_{M,X}\otimes_{\mathcal{H}} K_{M,X}\right] + 6\lambda(1 + \|g_\rho\|^2_{L_2(\rho_X)} + 24\|\Sigma_\infty^{-r}g_\rho\|^2_{L_2(\rho_X)})\Sigma_M,
$$

where we used $\Sigma_M(\Sigma_M + \lambda I)^{-1} \preccurlyeq I$ and $\lambda(\Sigma_M + \lambda I)^{-1} \preccurlyeq I$. Moreover, we see

$$
\mathbb{E}\left[K_{M,X}\otimes_{\mathcal{H}} K_{M,X}CH^{-1}K_{M,X}\otimes_{\mathcal{H}} K_{M,X}\right]
$$

$$
= 2(1 + \|g_\rho\|^2_{L_2(\rho_X)} + 24\|\Sigma_\infty^{-r}g_\rho\|^2_{L_2(\rho_X)})\mathbb{E}\left[K_{M,X}\otimes_{\mathcal{H}} K_{M,X}\Sigma_M(\Sigma_M + \lambda I)^{-1}K_{M,X}\otimes_{\mathcal{H}} K_{M,X}\right]
$$

$$
\preccurlyeq 2(1 + \|g_\rho\|^2_{L_2(\rho_X)} + 24\|\Sigma_\infty^{-r}g_\rho\|^2_{L_2(\rho_X)})\mathbb{E}\left[(K_{M,X}\otimes_{\mathcal{H}} K_{M,X})^2\right]
$$

$$
\preccurlyeq 24(1 + \|g_\rho\|^2_{L_2(\rho_X)} + 24\|\Sigma_\infty^{-r}g_\rho\|^2_{L_2(\rho_X)})\Sigma_M,
$$

where we used (27) for the last inequality. Hence, we get

$$
\mathbb{E}\left[(K_{M,X}\otimes_{\mathcal{H}} K_{M,X} + \lambda I)CH^{-1}(K_{M,X}\otimes_{\mathcal{H}} K_{M,X} + \lambda I)\right]
$$

$$
\preccurlyeq (24 + 6\lambda)(1 + \|g_\rho\|^2_{L_2(\rho_X)} + 24\|\Sigma_\infty^{-r}g_\rho\|^2_{L_2(\rho_X)})\Sigma_M.
$$

Since, $4\eta(6 + \lambda) \leq 1$, the condition (32) is verified. We apply Lemma 21 in Pillaud-Vivien et al. (2018a) to (31), yielding the following inequality:

$$
\mathbb{E}[\|\overline{A}_T\|^2_{L_2(\rho_X)}] \leq \frac{4}{T+1}\left(1 + \|g_\rho\|^2_{L_2(\rho_X)} + 24\|\Sigma_\infty^{-r}g_\rho\|^2_{L_2(\rho_X)}\right)\mathrm{Tr}\left(\Sigma_M^2(\Sigma_M + \lambda I)^{-2}\right)
$$

$$
\leq \frac{4}{T+1}\left(1 + \|g_\rho\|^2_{L_2(\rho_X)} + 24\|\Sigma_\infty^{-r}g_\rho\|^2_{L_2(\rho_X)}\right)\mathrm{Tr}\left(\Sigma_M(\Sigma_M + \lambda I)^{-1}\right). \tag{33}
$$

**Convergence rate in terms of the optimization.** Finally, by combining (30) and (33) with (22), we get the convergence rate of averaged stochastic gradient descent to $g_{M,\lambda}$:

$$
\mathbb{E}\left[\left\|\overline{g}^{(T)} - g_{M,\lambda}\right\|^2_{L_2(\rho_X)}\right] \leq \frac{4}{\eta^2(T+1)^2}\|(\Sigma_M + \lambda I)^{-1}g_{M,\lambda}\|^2_{L_2(\rho_X)}
$$

$$
+ \frac{2\cdot 24^2}{T+1}\|(\Sigma_M + \lambda I)^{-1/2}g_{M,\lambda}\|^2_{L_2(\rho_X)}
$$

$$
+ \frac{8}{T+1}\left(1 + \|g_\rho\|^2_{L_2(\rho_X)} + 24\|\Sigma_\infty^{-r}g_\rho\|^2_{L_2(\rho_X)}\right)\mathrm{Tr}\left(\Sigma_M(\Sigma_M + \lambda I)^{-1}\right),
$$

where $\overline{g}^{(T)} \stackrel{def}{=} \frac{1}{T+1}\sum_{t=0}^{T} g^{(t)}$. This finishes the proof. $\square$

## D    PROOF OF PROPOSITION B

We provide Proposition B which provides the bound on Theorem A.

*Proof of Proposition B.* From the Bernstein's inequality (Proposition 3 in Rudi & Rosasco (2017)) to random operators, the covariance operator $\Sigma_M$ converges to $\Sigma_\infty$ as $M \to \infty$ in probability. Especially, there exits $M_0 \in \mathbb{Z}_+$ such that for any $M \geq M_0$, it follows that with high probability at least $1 - \delta$, $\Sigma_\infty - \Sigma_M \preccurlyeq \frac{1}{2}(\Sigma_\infty + \lambda I)$ in $L_2(\rho_X)$. Thus, for $\forall f \in L_2(\rho_X)$, we see

$$\left\langle (\Sigma_\infty + \lambda I)^{-1/2}(\Sigma_\infty - \Sigma_M)(\Sigma_\infty + \lambda I)^{-1/2}f, f \right\rangle_{L_2(\rho_X)}$$
$$= \left\langle (\Sigma_\infty - \Sigma_M)(\Sigma_\infty + \lambda I)^{-1/2}f, (\Sigma_\infty + \lambda I)^{-1/2}f \right\rangle_{L_2(\rho_X)}$$
$$\leq \frac{1}{2} \left\langle (\Sigma_\infty + \lambda I)(\Sigma_\infty + \lambda I)^{-1/2}f, (\Sigma_\infty + \lambda I)^{-1/2}f \right\rangle_{L_2(\rho_X)}$$
$$= \frac{1}{2}\|f\|^2_{L_2(\rho_X)}.$$

Hence, we have

$$(\Sigma_\infty + \lambda I)^{-1/2}(\Sigma_\infty - \Sigma_M)(\Sigma_\infty + \lambda I)^{-1/2} \preccurlyeq \frac{1}{2}I.$$

Following the argument in Bach (2017b), we have for $\forall f \in L_2(\rho_X)$,

$$\left\langle (\Sigma_M + \lambda I)^{-1}f, f \right\rangle_{L_2(\rho_X)}$$
$$= \left\langle (\Sigma_\infty + \lambda I + \Sigma_M - \Sigma_\infty)^{-1}f, f \right\rangle_{L_2(\rho_X)}$$
$$= \left\langle \left(I + (\Sigma_\infty + \lambda I)^{-1/2}(\Sigma_M - \Sigma_\infty)(\Sigma_\infty + \lambda I)^{-1/2}\right)^{-1}(\Sigma_\infty + \lambda I)^{-1/2}f, (\Sigma_\infty + \lambda I)^{-1/2}f \right\rangle_{L_2(\rho_X)}$$
$$= 2\left\langle (\Sigma_\infty + \lambda I)^{-1/2}f, (\Sigma_\infty + \lambda I)^{-1/2}f \right\rangle_{L_2(\rho_X)}$$
$$= 2\left\langle (\Sigma_\infty + \lambda I)^{-1}f, f \right\rangle_{L_2(\rho_X)}.$$

Thus, we confirm that with high probability at least $1 - \delta$,

$$(\Sigma_M + \lambda I)^{-1} \preccurlyeq 2(\Sigma_\infty + \lambda I)^{-1} \tag{34}$$

Utilizing this inequality, we show the first and second inequalities in Proposition B as follows. It is sufficient to prove the second inequality because of

$$\|(\Sigma_M + \lambda I)^{-1}g_{M,\lambda}\|^2_{L_2(\rho_X)} \leq \frac{1}{\lambda}\|(\Sigma_M + \lambda I)^{-1/2}g_{M,\lambda}\|^2_{L_2(\rho_X)}$$

Noting that $g_\rho \in \mathcal{H}_\infty$ and $g_{M,\lambda} = (\Sigma_M + \lambda I)^{-1}\Sigma_M g_\rho$, we get

$$\|(\Sigma_M + \lambda I)^{-1/2}g_{M,\lambda}\|^2_{L_2(\rho_X)} = \|(\Sigma_M + \lambda I)^{-1/2}(\Sigma_M + \lambda I)^{-1}\Sigma_M g_\rho\|^2_{L_2(\rho_X)}$$
$$\leq \|(\Sigma_M + \lambda I)^{-1/2}g_\rho\|^2_{L_2(\rho_X)}$$
$$\leq 2\|(\Sigma_\infty + \lambda I)^{-1/2}g_\rho\|^2_{L_2(\rho_X)}$$
$$\leq 2\|\Sigma_\infty^{-1/2}g_\rho\|^2_{L_2(\rho_X)}$$
$$= 2\|g_\rho\|^2_{\mathcal{H}_\infty}.$$

The third inequality on the degree of freedom is a result obtained by Rudi & Rosasco (2017).    □

## E    EIGENVALUE ANALYSIS OF NEURAL TANGENT KERNEL

### E.1    REVIEW OF SPHERICAL HARMONICS

We briefly review the spherical harmonics which is useful in analyzing the eigenvalues of dot-product kernels. For references, see Atkinson & Han (2012); Bach (2017a); Bietti & Mairal (2019); Cao et al. (2019).

Here, we denote by $\tau_{d-1}$ is the uniform distribution on the sphere $\mathbb{S}^{d-1} \subset \mathbb{R}^d$. The surface area of $\mathbb{S}^{d-1}$ is $\omega_{d-1} = \frac{2\pi^{d/2}}{\Gamma(d/2)}$ where $\Gamma$ is the Gamma function. In $L_2(\tau_{d-1})$, there is an orthonomal basis consisting of a constant 1 and the *spherical harmonics* $Y_{kj}(x)$, $k \in \mathbb{Z}_{\geq 1}$, $j = 1, \ldots, N(d, k)$, where $N(d, k) = \frac{2k+d-2}{k} \begin{pmatrix} k + d - 3 \\ d - 2 \end{pmatrix}$. That is, $\langle Y_{ki}, Y_{sj} \rangle_{L_2(\tau_{d-1})} = \delta_{ks}\delta_{ij}$ and $\langle Y_{ki}, 1 \rangle_{L_2(\tau_{d-1})} = 0$. The spherical harmonics $Y_{kj}$ are homogeneous functions of degree $k$, and clearly $Y_{kj}$ have the same parity as $k$.

*Legendre* polynomial $P_k(t)$ of degree $k$ and dimension $d$ (a.k.a. *Gegenbauer polynomial*) is defined as (Rodrigues' formula):

$$P_k(t) = (-1/2)^k \frac{\Gamma(\frac{d-1}{2})}{\Gamma\left(k + \frac{d-1}{2}\right)} (1 - t^2)^{(3-d)/2} \left(\frac{\mathrm{d}}{\mathrm{d}t}\right)^k (1 - t^2)^{k+(d-3)/2}.$$

Legendre polynomials have the same parity as $k$. This polynomial is very useful in describing several formulas regarding the spherical harmonics.

**Addition formula.** We have the following addition formula:

$$\sum_{j=1}^{N(d,k)} Y_{kj}(x)Y_{kj}(y) = N(d, k)P_k(x^\top y), \quad \forall x, \forall y \in \mathbb{S}^{d-1}. \tag{35}$$

Hence, we see that $P_k(x^\top \cdot)$ is spherical harmonics of degree $k$. Using the addition formula and the orthogonality of spherical harmonics, we have

$$\int_{\mathbb{S}^{d-1}} P_j(Z^\top x)P_k(Z^\top y)\mathrm{d}\tau_{d-1}(Z) = \frac{\delta_{jk}}{N(d, k)}P_k(x^\top y). \tag{36}$$

Combining the following equation: for $x = te_d + \sqrt{1 - t^2}x'$, ($x \in \mathbb{S}^{d-1}$, $x' \in \mathbb{S}^{d-2}$, $t \in [-1, 1]$),

$$\frac{\omega_{d-1}}{\omega_{d-2}}\mathrm{d}\tau_{d-1}(x) = (1 - t^2)^{(d-3)/2}\mathrm{d}t\mathrm{d}\tau_{d-2}(x'),$$

we see the orthogonality of Legendre polynomials in $L_2([-1, 1], (1 - t^2)^{(d-3)/2}\mathrm{d}t)$ and since $P_k(1) = 1$ we see

$$\int_{-1}^{1} P_k^2(t)(1 - t^2)^{(d-3)/2}\mathrm{d}t = \frac{\omega_{d-1}}{\omega_{d-2}} \frac{1}{N(d, k)}.$$

**Recurrence relation.** We have the following relation:

$$tP_k(t) = \frac{k}{2k + d - 2}P_{k-1}(t) + \frac{k + d - 2}{2k + d - 2}P_{k+1}(t), \tag{37}$$

for $k \geq 1$, and for $k = 0$ we have $tP_0(t) = P_1(t)$.

**Funk-Hecke formula.** The following formula is useful in computing Fourier coefficients with respect to spherical harmonics via Legendre polynomials. For any linear combination $Y_k$ of $Y_{kj}$, ($j \in \{1, \ldots, N(d, k)\}$) and any $f \in L_2([-1, 1], (1 - t^2)^{(d-3)/2}\mathrm{d}t)$, we have for $\forall x$,

$$\int_{\mathbb{S}^{d-1}} f(x^\top y)Y_k(y)\mathrm{d}\tau_{d-1}(y) = \frac{\omega_{d-2}}{\omega_{d-1}}Y_k(x) \int_{-1}^{1} f(t)P_k(t)(1 - t^2)^{(d-3)/2}\mathrm{d}t. \tag{38}$$

This formula say that spherical harmonics are eigenfunctions of the integral operator defined by $f(x^\top y)$ and each eigen-space is spanned by spherical harmonics of the same degree. Moreover, it also provides a way of computing corresponding eigenvalues.

## E.2 Eigenvalues of Dot-product Kernels

Let $\mu_0$ be the uniform distribution on $\mathbb{S}^{d-1}$. Note that although $\tau_{d-1}$ and $\mu_0$ are the same distribution, we use two distributions $\tau_{d-1}$ and $\mu_0$ depending on random variables. First, we consider any activation function $\sigma : \mathbb{R} \to \mathbb{R}$ and a kernel function $k(x, x') = \mathbb{E}_{b^{(0)} \sim \mu_0}[\sigma(b^{(0)\top} x)\sigma(b^{(0)\top} x')]$ on the sphere $\mathbb{S}^{d-1}$. We show this kernel function is a type of dot-product kernels, that is, there is $\hat{k} : \mathbb{R} \to \mathbb{R}$ such that $k(x, x') = \hat{k}(x^\top x')$. In fact, it can be confirmed as follows. For any $x, x' \in \mathbb{S}^{d-1}$, we take $\theta \in [0, \pi]$ so that $x^\top x' = \cos\theta$, and an orthogonal matrix $A \in \mathbb{R}^{d \times d}$ so that $Ax = (1, 0, \ldots, 0)^\top$ and $Ax' = (\cos\theta, \sin\theta, 0, \ldots, 0)^\top$ because $A$ preserves the value of $x^\top x'$. Then, since $\mu_0$ is rotationally invariant we see

$$k(x, x') = \int_{\mathbb{S}^{d-1}} \sigma(b^{(0)\top} Ax)\sigma(b^{(0)\top} Ax')\mathrm{d}\mu_0(b^{(0)})$$

$$= \int_{\mathbb{S}^{d-1}} \sigma(b_1^{(0)})\sigma(b_1^{(0)}\cos(\theta) + b_2^{(0)}\sin(\theta))\mathrm{d}\mu_0(b^{(0)}),$$

where $b^{(0)} = (b_1^0, b_2^{(0)}, \ldots, b_d^{(0)})$. In other words, we see $k$ is a function of $\theta = \arccos(x^\top x')$, and is a dot-product kernel $k(x, x') = \hat{k}(x^\top x')$. Hence, we can apply Funk-Hecke formula (38) to $k(x, \cdot)$.

The derivation of eigenvalues of the integral operator follows a way developed by Bach (2017a); Bietti & Mairal (2019); Cao et al. (2019). In general, $g \in L_2(\tau_{d-1})$ can be decomposed by spherical harmonics as follows.

$$g - \int_{\mathbb{S}^{d-1}} g(Z)\mathrm{d}\tau_{d-1}(Z) = \sum_{k=1}^{\infty} \sum_{j=1}^{N(d,k)} \langle g, Y_{kj}\rangle_{L_2(\tau_{d-1})} Y_{kj}$$

$$= \sum_{k=1}^{\infty} \sum_{j=1}^{N(d,k)} \int_{\mathbb{S}^{d-1}} g(Z)Y_{kj}(Z)Y_{kj}(\cdot)\mathrm{d}\tau_{d-1}(Z)$$

$$= \sum_{k=1}^{\infty} N(d,k) \int_{\mathbb{S}^{d-1}} g(Z)P_k(Z^\top \cdot)\mathrm{d}\tau_{d-1}(Z), \tag{39}$$

where we used addition formula to the last equality.

Here, we apply this decomposition (39) to $k(x, \cdot) = \hat{k}(x^\top \cdot)$. Since $P_k(Z^\top \cdot)$ is a linear combination of spherical harmonics of degree $k$ (see addition formula), we get

$$k(x, \cdot) - \int_{\mathbb{S}^{d-1}} \hat{k}(x^\top Z)\mathrm{d}\tau_{d-1}(Z) = \sum_{k=1}^{\infty} N(d,k) \int_{\mathbb{S}^{d-1}} \hat{k}(x^\top Z)P_k(Z^\top \cdot)\mathrm{d}\tau_{d-1}(Z)$$

$$= \sum_{k=1}^{\infty} \hat{\lambda}_k N(d,k)P_k(x^\top \cdot), \tag{40}$$

where we used Funk-Hecke formula (38) and we set $\hat{\lambda}_k = \frac{\omega_{d-2}}{\omega_{d-1}} \int_{-1}^{1} \hat{k}(t)P_k(t)(1-t^2)^{(d-3)/2}\mathrm{d}t$. We note that $\hat{\lambda}_k$ is eigenvalue with multiplicity $N(d,k)$ of the integral operator defined by $k$.

Next, we derive another expression of $k$. In a similar way, we obtain the following equation:

$$\sigma(b^{(0)\top} x) - \int_{\mathbb{S}^{d-1}} \sigma(Z^\top x)\mathrm{d}\tau_{d-1}(Z) = \sum_{k=1}^{\infty} \hat{\mu}_k N(d,k)P_k(b^{(0)\top} x),$$

where $\hat{\mu}_k = \frac{\omega_{d-2}}{\omega_{d-1}} \int_{-1}^{1} \sigma(t)P_k(t)(1-t^2)^{(d-3)/2}\mathrm{d}t$. By the definition of $k$ and the orthogonality of spherical harmonics, we get

$$k(x, x') = \mathbb{E}_{b^{(0)}} \left[\sigma(b^{(0)\top} x)\sigma(b^{(0)\top} x')\right]$$

$$= \int_{\mathbb{S}^{d-1}} \sigma(Z^\top x)\mathrm{d}\tau_{d-1}(Z) \int_{\mathbb{S}^{d-1}} \sigma(Z^\top x')\mathrm{d}\tau_{d-1}(Z) + \sum_{k=1}^{\infty} \hat{\mu}_k^2 N^2(d,k)\mathbb{E}_{b^{(0)}}\left[P_k(b^{(0)\top} x)P_k(b^{(0)\top} x')\right]$$

$$= \int_{\mathbb{S}^{d-1}} \sigma(Z^\top x)\mathrm{d}\tau_{d-1}(Z) \int_{\mathbb{S}^{d-1}} \sigma(Z^\top x')\mathrm{d}\tau_{d-1}(Z) + \sum_{k=1}^{\infty} \hat{\mu}_k^2 N(d,k)P_k(x^\top x'), \tag{41}$$

where we used equation (36). By the rotationally invariance, we can show

$$\int_{\mathbb{S}^{d-1}} \hat{k}(x^\top Z)\mathrm{d}\tau_{d-1}(Z) = \int_{\mathbb{S}^{d-1}} \sigma(Z^\top x)\mathrm{d}\tau_{d-1}(Z) \int_{\mathbb{S}^{d-1}} \sigma(Z^\top x')\mathrm{d}\tau_{d-1}(Z).$$

Thus, comparing (40) with (41), we get $\hat{\lambda}_k = \hat{\mu}_k^2$.

### E. 3   EIGENVALUES OF NEURAL TANGENT KERNELS

Utilizing a relation $\hat{\lambda}_k = \hat{\mu}_k^2$, we derive a way of computing eigenvalues of the integral operator defined by the integral operators $\Sigma_\infty$ associated with the activation $\sigma$. Recall the definition of the neural tangent kernel:

$$k_\infty(x, x') \stackrel{def}{=} \mathbb{E}_{b^{(0)} \sim \mu_0}[\sigma(b^{(0)\top}x)\sigma(b^{(0)\top}x')] + (x^\top x' + \gamma^2)\mathbb{E}_{b^{(0)} \sim \mu_0}[\sigma'(b^{(0)\top}x)\sigma'(b^{(0)\top}x')].$$

A neural tangent kernel consists of three kernels:

$$h_1(x, x') = \mathbb{E}_{b^{(0)} \sim \mu_0}\left[\sigma(b^{(0)\top}x)\sigma(b^{(0)\top}x')\right],$$
$$h_2(x, x') = \mathbb{E}_{b^{(0)} \sim \mu_0}\left[\sigma'(b^{(0)\top}x)\sigma'(b^{(0)\top}x')\right],$$
$$h_3(x, x') = x^\top x'\mathbb{E}_{b^{(0)} \sim \mu_0}\left[\sigma'(b^{(0)\top}x)\sigma'(b^{(0)\top}x')\right].$$

By the argument in the previous subsection, $h_1$ and $h_2$ are dot-product kernel, that is, there exist $\hat{h}_1$ and $\hat{h}_2$ such that $h_1(x, x') = \hat{h}_1(x^\top x')$ and $h_2(x, x') = \hat{h}_2(x^\top x')$. Moreover, $h_3$ is a dot-product kernel as well because we get $h_3(x, x') = \hat{h}_3(x^\top x')$ by setting $\hat{h}_3(t) = t\hat{h}_2(t)$. Hence, theory explained earlier is applicable to these kernels.

Eigenvalues $\hat{\mu}_k$ for kernels $h_1$ and $h_2$ are described as follows:

$$\hat{\mu}_k^{(1)} = \frac{\omega_{d-2}}{\omega_{d-1}} \int_{-1}^1 \sigma(t)P_k(t)(1 - t^2)^{(d-3)/2}\mathrm{d}t, \tag{42}$$

$$\hat{\mu}_k^{(2)} = \frac{\omega_{d-2}}{\omega_{d-1}} \int_{-1}^1 \sigma'(t)P_k(t)(1 - t^2)^{(d-3)/2}\mathrm{d}t, \tag{43}$$

yielding eigenvalues $\hat{\lambda}_k^{(1)} = (\hat{\mu}_k^{(1)})^2$ and $\hat{\lambda}_k^{(2)} = (\hat{\mu}_k^{(2)})^2$ for $h_1$ and $h_2$, respectively. As for eigenvalues $\hat{\lambda}_k^{(3)}$ for $h_3$, we have

$$\begin{aligned}
\hat{\lambda}_k^{(3)} &= \frac{\omega_{d-2}}{\omega_{d-1}} \int_{-1}^1 t\hat{h}_2(t)P_k(t)(1 - t^2)^{(d-3)/2}\mathrm{d}t \\
&= \frac{k}{2k + d - 2}\frac{\omega_{d-2}}{\omega_{d-1}} \int_{-1}^1 \hat{h}_2(t)P_{k-1}(t)(1 - t^2)^{(d-3)/2}\mathrm{d}t \\
&\quad + \frac{k + d - 2}{2k + d - 2}\frac{\omega_{d-2}}{\omega_{d-1}} \int_{-1}^1 \hat{h}_2(t)P_{k+1}(t)(1 - t^2)^{(d-3)/2}\mathrm{d}t \\
&= \frac{k}{2k + d - 2}\hat{\lambda}_{k-1}^{(2)} + \frac{k + d - 2}{2k + d - 2}\hat{\lambda}_{k+1}^{(2)},
\end{aligned}$$

where we used the recurrence relation (37). Since, $h_1$, $h_2$, and $h_3$ have the same eigenfunctions, eigenvalues $\hat{\lambda}_{\infty,k}$ of $k_\infty$ is

$$\hat{\lambda}_{\infty,k} = \hat{\lambda}_k^{(1)} + \gamma^2\hat{\lambda}_k^{(2)} + \frac{k}{2k + d - 2}\hat{\lambda}_{k-1}^{(2)} + \frac{k + d - 2}{2k + d - 2}\hat{\lambda}_{k+1}^{(2)}. \tag{44}$$

Hence, calculation of $\{\hat{\lambda}_{\infty,k}\}_{k=1}^\infty$ results in computing $\hat{\mu}_k^{(1)}$ and $\hat{\mu}_k^{(2)}$ for given activation $\sigma$.

**Eigenvalues for ReLU and smooth approximations of ReLU.** As for ReLU activation, its eigenvalues were derived in Bach (2017a). Let $\sigma$ be ReLU. Then, $\hat{\mu}_k^{(1)} = 0$ and $\hat{\mu}_k^{(2)} \sim k^{-d/2}$ when $k$ is odd and $\hat{\mu}_k^{(1)} \sim k^{-d/2-1}$ and $\hat{\mu}_k^{(2)} = 0$ when $k$ is even. Consequently, we see $\hat{\lambda}_{\infty,k} = \Theta(k^{-d})$.

We note that the multiplicity of $\hat{\lambda}_{\infty,k}$ is $N(d,k)$, so that we should take into account the multiplicity to derive decay order of eigenvalues $\lambda_{\infty,i}$ of $\Sigma_\infty$. Since $1 + \sum_{j=1}^k N(d-1,j) = N(d,k)$ (for details see Atkinson & Han (2012)), we see $\lambda_{\infty,N(d+1,k)} = \Theta(k^{-d})$. Moreover, $N(d+1,k) = \Theta(k^{d-1})$ yields $\lambda_{\infty,N(d+1,k)} = \Theta\left(N(d+1,k)^{-1-\frac{1}{d-1}}\right)$. As a result, Assumption **(A4)** is verified with $\beta = 1 + \frac{1}{d-1}$ for ReLU.

For the smooth approximation $\sigma^{(s)}$ of ReLU satisfying Assumption **(A1')**, we can show that every eigenvalue of $\Sigma_\infty^{(s)}$ derived from $\sigma^{(s)}$ converges to that for ReLU as $s \to \infty$ because of (42) and (43) with Lebesgue's convergence theorem.

## F    EXPLICIT CONVERGENCE RATES FOR SMOOTH APPROXIMATION OF RELU

For convenience, we here list notations used in this section. In this section, let $\sigma$ and $\sigma^{(s)}$ be ReLU activation and its smooth approximation satisfying **(A1')**, respectively, and for $M \in \mathbb{Z}_+ \cup \{\infty\}$ let $k_M, \Sigma_M, g_{M,\lambda}, k_M^{(s)}, \Sigma_M^{(s)}, g_{M,\lambda}^{(s)}$ be corresponding kernel, integral operators, and minimizers of the regularized expected risk functions. Let $\overline{g}^{(T)}$ be iterates obtained by the reference ASGD (Algorithm 2) in the RKHS associated with $k_M^{(s)}$.

We consider the following decomposition:

$$\frac{1}{3}\|\overline{g}^{(T)} - g_\rho\|_{L_2(\rho_X)}^2 \le \|\overline{g}^{(T)} - g_{M,\lambda}^{(s)}\|_{L_2(\rho_X)}^2 \tag{45}$$

$$+ \|g_{M,\lambda}^{(s)} - g_{\infty,\lambda}^{(s)}\|_{L_2(\rho_X)}^2 \tag{46}$$

$$+ \|g_{\infty,\lambda}^{(s)} - g_{\infty,\lambda}\|_{L_2(\rho_X)}^2 \tag{47}$$

$$+ \|g_{\infty,\lambda} - g_\rho\|_{L_2(\rho_X)}^2. \tag{48}$$

These terms can be made arbitrarily small by taking large $M$ and $s$. As for (48) this property is a direct consequence of Proposition D. Note that Proposition C is not applicable to (46) because this proposition require the specification of the target function by $k_\infty^{(s)}$ which does not hold in general.

In the following, we treat the remaining terms.

**Proposition E.** *Suppose* **(A1')** *and* **(A2')** *hold. Then, we have*

1. $\mathrm{plim}_{M\to\infty}\|k_M^{(s)} - k_\infty^{(s)}\|_{L_\infty(\rho_X)^2} = 0$, $\lim_{s\to\infty}\|k_\infty^{(s)} - k_\infty\|_{L_\infty(\rho_X)^2} = 0$,

2. $\mathrm{plim}_{M\to\infty}\left|\mathrm{Tr}\left(\Sigma_M^{(s)} - \Sigma_\infty^{(s)}\right)\right| = 0$, $\lim_{s\to\infty}\left|\mathrm{Tr}\left(\Sigma_\infty^{(s)} - \Sigma_\infty\right)\right| = 0$,

3. $\mathrm{plim}_{M\to\infty}\|g_{M,\lambda}^{(s)} - g_{\infty,\lambda}^{(s)}\|_{L_\infty(\rho_X)} = 0$, $\lim_{s\to\infty}\|g_{\infty,\lambda}^{(s)} - g_{\infty,\lambda}\|_{L_\infty(\rho_X)} = 0$,

*where* $\mathrm{plim}$ *denotes the convergence in probability.*

*Proof.* We show the first statement. By the uniform law of large numbers (Theorem 3.1 in Mohri et al. (2012)), we see the convergence in probability:

$$\|k_M^{(s)} - k_\infty^{(s)}\|_{L_\infty(\rho_X)^2} \le \sup_{x,x'\in\mathbb{S}^{d-1}} \left|\frac{1}{M}\sum_{r=1}^M \sigma^{(s)}(b_r^{(0)\top}x)\sigma^{(s)}(b_r^{(0)\top}x') - \mathbb{E}_{b^{(0)}}\left[\sigma^{(s)}(b^{(0)\top}x)\sigma^{(s)}(b^{(0)\top}x')\right]\right|$$

$$+ (1+\gamma^2)\sup_{x,x'\in\mathbb{S}^{d-1}}\left|\frac{1}{M}\sum_{r=1}^M \sigma^{(s)\prime}(b_r^{(0)\top}x)\sigma^{(s)\prime}(b_r^{(0)\top}x') - \mathbb{E}_{b^{(0)}}\left[\sigma^{(s)\prime}(b^{(0)\top}x)\sigma^{(s)\prime}(b^{(0)\top}x')\right]\right| \xrightarrow{p} 0,$$

where the limit is taken with respect to $M \to \infty$ and the notation $\xrightarrow{p}$ denotes the convergence in probability. Next, we have the following convergence:

$$\|k_\infty^{(s)} - k_\infty\|_{L_\infty(\rho_X)^2} \leq \sup_{x,x'\in\mathbb{S}^{d-1}} \mathbb{E}_{b^{(0)}}\left[\left|\sigma^{(s)}(b^{(0)\top}x)\sigma^{(s)}(b^{(0)\top}x') - \sigma(b^{(0)\top}x)\sigma(b^{(0)\top}x')\right|\right]$$

$$+ (1+\gamma^2)\sup_{x,x'\in\mathbb{S}^{d-1}} \mathbb{E}_{b^{(0)}}\left[\left|\sigma^{(s)'}(b^{(0)\top}x)\sigma^{(s)'}(b^{(0)\top}x') - \sigma'(b^{(0)\top}x)\sigma'(b^{(0)\top}x')\right|\right]$$

$$\leq 4\sup_{x\in\mathbb{S}^{d-1}}\mathbb{E}_{b^{(0)}}\left[\left|\sigma^{(s)}(b^{(0)\top}x) - \sigma(b^{(0)\top}x)\right|\right] + 4(1+\gamma^2)\sup_{x\in\mathbb{S}^{d-1}}\mathbb{E}_{b^{(0)}}\left[\left|\sigma^{(s)'}(b^{(0)\top}x) - \sigma'(b^{(0)\top}x)\right|\right]$$

$$= 4\left[\left|\sigma^{(s)}(b^{(0)\top}e_1) - \sigma(b^{(0)\top}e_1)\right|\right] + 4(1+\gamma^2)\mathbb{E}_{b^{(0)}}\left|\sigma^{(s)'}(b^{(0)\top}e_1) - \sigma'(b^{(0)\top}e_1)\right| \to 0,$$

where for the first inequality we used the boundedness of $\sigma, \sigma', \sigma^{(s)}$, and $\sigma^{(s)'}$ on $[-1,1]$, for the equality we used the rotationally invariance of the measure $\mu_0$, and the limit is taken with respect to $s \to \infty$. For the final convergence in the above expression we used Assumption **(A1')** and boundedness with Lebesgue's convergence theorem.

In general, for a kernel $k$ and associated integral operator $\Sigma$ with $\rho_X$, we have $\mathrm{Tr}\,(\Sigma) = \int_{\mathbb{S}^{d-1}} k(X,X)\mathrm{d}\rho_X$. Hence, $\left|\mathrm{Tr}\left(\Sigma_M^{(s)} - \Sigma_\infty^{(s)}\right)\right| \leq \|k_M^{(s)} - k_\infty^{(s)}\|_{L_\infty(\rho_X)^2}$ and $\left|\mathrm{Tr}\left(\Sigma_\infty^{(s)} - \Sigma_\infty\right)\right| \leq \|k_\infty^{(s)} - k_\infty\|_{L_\infty(\rho_X)^2}$, and the second statement holds immediately.

Finally, we show the third statement. In the same manner as the derivation of inequality (19), we get

$$\|((\Sigma_M^{(s)} + \lambda I)^{-1} - (\Sigma_\infty^{(s)} + \lambda I)^{-1})g_\rho\|_{L_2(\rho_X)} \leq \frac{1}{\lambda^2}\|\Sigma_M^{(s)} - \Sigma_\infty^{(s)}\|_{\mathrm{op}}\|g_\rho\|_{L_2(\rho_X)}.$$

We denote $F_M^{(s)} = (\Sigma_M^{(s)} + \lambda I)^{-1}g_\rho$ and $F_\infty^{(s)} = (\Sigma_\infty^{(s)} + \lambda I)^{-1}g_\rho$. Noting $g_{M,\lambda}^{(s)} = \Sigma_M^{(s)}F_M^{(s)}$ and $g_{\infty,\lambda}^{(s)} = \Sigma_\infty^{(s)}F_\infty^{(s)}$, we get for $x \in \mathbb{S}^{d-1}$,

$$|g_{M,\lambda}^{(s)}(x) - g_{\infty,\lambda}^{(s)}(x)|$$

$$= \left|\Sigma_M^{(s)}F_M^{(s)}(x) - \Sigma_\infty^{(s)}F_\infty^{(s)}(x)\right|$$

$$= \left|\int_\mathcal{X} K_{M,x}^{(s)}(X)F_M^{(s)}(X)\mathrm{d}\rho_X - \int_\mathcal{X} K_{\infty,x}^{(s)}(X')F_\infty^{(s)}(X')\mathrm{d}\rho_X\right|$$

$$= \left|\int_\mathcal{X}(K_{M,x}^{(s)} - K_{\infty,x}^{(s)})(X)F_M^{(s)}(X)\mathrm{d}\rho_X - \int_\mathcal{X} K_{\infty,x}^{(s)}(X)(F_M^{(s)} - F_\infty^{(s)})(X)\mathrm{d}\rho_X\right|$$

$$\leq \|K_{M,x}^{(s)} - K_{\infty,x}^{(s)}\|_{L_2(\rho_X)}\|F_M^{(s)}\|_{L_2(\rho_X)} + \|K_{\infty,x}^{(s)}\|_{L_2(\rho_X)}\|F_M^{(s)} - F_\infty^{(s)}\|_{L_2(\rho_X)}$$

$$\leq \frac{1}{\lambda}\|k_M^{(s)} - k_\infty^{(s)}\|_{L_\infty(\rho_X)^2}\|g_\rho\|_{L_2(\rho_X)} + \frac{12}{\lambda^2}\|\Sigma_M^{(s)} - \Sigma_\infty^{(s)}\|_{\mathrm{op}}\|g_\rho\|_{L_2(\rho_X)}. \tag{49}$$

The both terms in the last expression (49) converge to 0 in probability because of the first statement of this proposition and the Bernstein's inequality (Proposition 3 in Rudi & Rosasco (2017)) to random operators. This finishes the proof of the former of the third statement.

In the same manner, we have

$$\|g_{\infty,\lambda}^{(s)} - g_{\infty,\lambda}\|_{L_\infty(\rho_X)} \leq \frac{1}{\lambda}\|k_\infty - k_\infty^{(s)}\|_{L_\infty(\rho_X)^2}\|g_\rho\|_{L_2(\rho_X)} + \frac{12}{\lambda^2}\|\Sigma_\infty - \Sigma_\infty^{(s)}\|_{\mathrm{op}}\|g_\rho\|_{L_2(\rho_X)}. \tag{50}$$

The first term in the right hand side converges to 0 because of the first statement of this proposition. We next show the convergence $\|\Sigma_\infty - \Sigma_\infty^{(s)}\|_{\mathrm{op}} \to 0$ as $s \to \infty$. As seen in the previous section, $\Sigma_\infty$ and $\Sigma_\infty^{(s)}$ share the same eigenfunctions and every eigenvalue of $\Sigma_\infty^{(s)}$ converges to that of $\Sigma_\infty$ as $s \to \infty$. Let $\{\lambda_{\infty,i}^{(s)}\}_{i=1}^\infty$ and $\{\lambda_{\infty,i}\}_{i=1}^\infty$ be eigenvalues of $\Sigma_\infty^{(s)}$ and $\Sigma_\infty$, respectively. For an arbitrary $\epsilon > 0$, we can take $i_\epsilon$ such that $\sum_{i=i_\epsilon}^\infty \lambda_{\infty,i} < \epsilon$. From the convergence $\lambda_{\infty,i}^{(s)} \to \lambda_{\infty,i}$ and $\mathrm{Tr}\left(\Sigma_\infty^{(s)}\right) \to \mathrm{Tr}\,(\Sigma_\infty)$ as $s \to \infty$, we see that for arbitrarily sufficiently large $s$, $|\lambda_{\infty,i}^{(s)} - \lambda_{\infty,i}| < \epsilon$ $(i < i_\epsilon)$ and $\sum_{i=i_\epsilon}^\infty \lambda_{\infty,i}^{(s)} < 2\epsilon$. Clearly, for $i \geq i_\epsilon$, $|\lambda_{\infty,i}^{(s)} - \lambda_{\infty,i}| \leq \sum_{i=i_\epsilon}^\infty(\lambda_{\infty,i}^{(s)} + \lambda_{\infty,i}) < 3\epsilon$.

Therefore, we conclude the uniform convergence $\sup_{i \in \{1,\ldots,\infty\}} |\lambda_{\infty,i}^{(s)} - \lambda_{\infty,i}| \to 0$ as $s \to \infty$. This implies $\|\Sigma_\infty - \Sigma_\infty^{(s)}\|_{\mathrm{op}} \to 0$ as $s \to \infty$. $\qquad\square$

So far, we have shown that (46), (47), and (48) can be made arbitrarily small by taking large $s$ and $M$ depending on $\lambda$. The remaining problem is to show the convergence of (45). To do so, we establish the counterpart of Theorem B by adapting Theorem A and Proposition B to the current setting.

**The counterpart of Theorem B.** In Theorem A, the condition **(A3)** is required for NTK associated with the smooth activation $\sigma^{(s)}$ and it is not satisfied in general. Note that **(A3)** is used for bounding $\|g_{M,\lambda}^{(s)}\|_{L_\infty(\rho_X)}$ uniformly as seen in the proof of Lemma A. Let us consider the decomposition:

$$
\begin{aligned}
\|g_{M,\lambda}^{(s)}\|_{L_\infty(\rho_X)} &\leq \|g_{\infty,\lambda}\|_{L_\infty(\rho_X)} + \|g_{M,\lambda}^{(s)} - g_{\infty,\lambda}^{(s)}\|_{L_\infty(\rho_X)} + \|g_{\infty,\lambda}^{(s)} - g_{\infty,\lambda}\|_{L_\infty(\rho_X)} \\
&\leq 2\sqrt{3}\|\Sigma_\infty^{-r} g_\rho\|_{L_2(\rho_X)} + \|g_{M,\lambda}^{(s)} - g_{\infty,\lambda}^{(s)}\|_{L_\infty(\rho_X)} + \|g_{\infty,\lambda}^{(s)} - g_{\infty,\lambda}\|_{L_\infty(\rho_X)} \\
&\to 2\sqrt{3}\|\Sigma_\infty^{-r} g_\rho\|_{L_2(\rho_X)}.
\end{aligned}
$$

Here, for the second inequality we used (20). Note that the inequality (20) holds for $\Sigma_\infty$ because the condition **(A3)** is supposed for ReLU. For the last inequality we used Proposition E. Hence, Theorem A can be applicable and the same convergence in Theorem A holds for $\sigma^{(s)}$. For arbitrarily sufficiently large $s$ and $M$ with high probability, we have

$$
\begin{aligned}
\mathbb{E}\left[\left\|\overline{g}^{(T)} - g_{M,\lambda}^{(s)}\right\|_{L_2(\rho_X)}^2\right] \leq{} & \frac{4}{\eta^2(T+1)^2}\|(\Sigma_M^{(s)} + \lambda I)^{-1} g_{M,\lambda}^{(s)}\|_{L_2(\rho_X)}^2 \\
& + \frac{2 \cdot 24^2}{T+1}\|(\Sigma_M^{(s)} + \lambda I)^{-1/2} g_{M,\lambda}^{(s)}\|_{L_2(\rho_X)}^2 \\
& + \frac{8}{T+1}\left(1 + \|g_\rho\|_{L_2(\rho_X)}^2 + 24\|\Sigma_\infty^{-r} g_\rho\|_{L_2(\rho_X)}^2\right) \mathrm{Tr}\left(\Sigma_M^{(s)}(\Sigma_M^{(s)} + \lambda I)^{-1}\right).
\end{aligned}
\tag{51}
$$

Next, we adapt Proposition B to the current setting. By inequality (34), there exists $M_0 \in \mathbb{Z}_+$ such that $\forall M \geq M_0$, with high probability,

$$
\|(\Sigma_M^{(s)} + \lambda I)^{-1/2} g_{M,\lambda}^{(s)}\|_{L_2(\rho_X)}^2 \leq 2\|(\Sigma_\infty^{(s)} + \lambda I)^{-1/2} g_\rho\|_{L_2(\rho_X)}^2.
$$

If $(\Sigma_\infty^{(s)} + \lambda I)^{-1} \preccurlyeq 2(\Sigma_\infty + \lambda I)^{-1}$ holds, then we have the counterpart of the second inequality in Proposition B because

$$
\|(\Sigma_\infty + \lambda I)^{-1/2} g_\rho\|_{L_2(\rho_X)}^2 \leq \|\Sigma_\infty^{-1/2} g_\rho\|_{L_2(\rho_X)}^2 = \|g_\rho\|_{\mathcal{H}_\infty}^2,
\tag{52}
$$

where we used the fact that $g_\rho$ is contained in $\mathcal{H}_\infty$ because of **(A3')**. Note that the first inequality in Proposition B is a direct consequence of the second one. We consider eigenvalues $\{\lambda_{\infty,i}^{(s)}\}_{i=1}^\infty$ and $\{\lambda_{\infty,i}\}_{i=1}^\infty$ of $\Sigma_\infty^{(s)}$ and $\Sigma_\infty$, respectively. Let $i_\lambda$ be an index such that for $\forall i > i_\lambda$, $\lambda_{\infty,i} \leq \frac{\lambda}{2}$. Since, every eigenvalue of $\{\lambda_{\infty,i}^{(s)}\}_{i=1}^\infty$ converges to that of $\{\lambda_{\infty,i}\}_{i=1}^\infty$ as $s \to \infty$, for an arbitrarily sufficiently large $s$, we have $|\lambda_{\infty,i}^{(s)} - \lambda_{\infty,i}| \leq \frac{\lambda}{2}$ for $\forall i < i_\lambda$, leading to $1/(\lambda + \lambda_{\infty,i}^{(s)}) \leq 2/(\lambda + \lambda_{\infty,i})$. As for the case $i \geq i_\lambda$, since $\frac{3}{2}\lambda \geq \lambda + \lambda_{\infty,i}$, we have $1/(\lambda + \lambda_{\infty,i}^{(s)}) \leq 1/\lambda \leq 3/(2(\lambda + \lambda_{\infty,i}))$. Combining these, we obtain $(\Sigma_\infty^{(s)} + \lambda I)^{-1} \preccurlyeq 2(\Sigma_\infty + \lambda I)^{-1}$ and

$$
\lim_{s \to \infty} \mathrm{plim}_{M \to \infty} \|(\Sigma_M^{(s)} + \lambda I)^{-1} g_{M,\lambda}^{(s)}\|_{L_2(\rho_X)}^2 \leq 4\lambda^{-1}\|g_\rho\|_{\mathcal{H}_\infty}^2,
\tag{53}
$$

$$
\lim_{s \to \infty} \mathrm{plim}_{M \to \infty} \|(\Sigma_M^{(s)} + \lambda I)^{-1/2} g_{M,\lambda}^{(s)}\|_{L_2(\rho_X)}^2 \leq 4\|g_\rho\|_{\mathcal{H}_\infty}^2.
\tag{54}
$$

These are the counterpart of the first and second inequalities in Proposition B.

Next, we consider the bound on the degree of freedom in this proposition. Assume $\lambda \leq \frac{1}{2}\|\Sigma_\infty\|_{\mathrm{op}}$. As seen earlier, an operator $\Sigma_\infty^{(s)}$ converge to $\Sigma_\infty$ in terms of the operator norm. Hence, $\lambda \leq \|\Sigma_\infty^{(s)}\|_{\mathrm{op}}$

for an arbitrarily sufficiently large $s$ and the bound on the degree of freedom in Proposition B is applicable. We get

$$\text{Tr}\left(\Sigma_M^{(s)}(\Sigma_M^{(s)} + \lambda I)^{-1}\right) \leq 3\text{Tr}\left(\Sigma_\infty^{(s)}(\Sigma_\infty^{(s)} + \lambda I)^{-1}\right). \tag{55}$$

Let us consider upper bounding the right hand side:

$$\begin{aligned}
\text{Tr}\left(\Sigma_\infty^{(s)}(\Sigma_\infty^{(s)} + \lambda I)^{-1}\right) &= \sum_{i=1}^{i_\lambda-1} \frac{\lambda_{\infty,i}^{(s)}}{\lambda + \lambda_{\infty,i}^{(s)}} + \sum_{i=i_\lambda}^{\infty} \frac{\lambda_{\infty,i}^{(s)}}{\lambda + \lambda_{\infty,i}^{(s)}} \\
&\leq \sum_{i=1}^{i_\lambda-1} \frac{\lambda_{\infty,i}^{(s)}}{\lambda + \lambda_{\infty,i}^{(s)}} + \frac{1}{\lambda}\sum_{i=i_\lambda}^{\infty} \lambda_{\infty,i}^{(s)} \\
&= \sum_{i=1}^{i_\lambda-1} \frac{\lambda_{\infty,i}^{(s)}}{\lambda + \lambda_{\infty,i}^{(s)}} - \frac{1}{\lambda}\sum_{i=1}^{i_\lambda-1} \lambda_{\infty,i}^{(s)} + \frac{1}{\lambda}\text{Tr}\left(\Sigma_\infty^{(s)}\right).
\end{aligned}$$

On the other hand, by the definition of $i_\lambda$,

$$\begin{aligned}
2\text{Tr}\left(\Sigma_\infty(\Sigma_\infty + \lambda I)^{-1}\right) &= 2\sum_{i=1}^{i_\lambda-1} \frac{\lambda_{\infty,i}}{\lambda + \lambda_{\infty,i}} + 2\sum_{i=i_\lambda}^{\infty} \frac{\lambda_{\infty,i}}{\lambda + \lambda_{\infty,i}} \\
&\geq 2\sum_{i=1}^{i_\lambda-1} \frac{\lambda_{\infty,i}}{\lambda + \lambda_{\infty,i}} + \frac{1}{\lambda}\sum_{i=i_\lambda}^{\infty} \lambda_{\infty,i} \\
&= 2\sum_{i=1}^{i_\lambda-1} \frac{\lambda_{\infty,i}}{\lambda + \lambda_{\infty,i}} - \frac{1}{\lambda}\sum_{i=1}^{i_\lambda-1} \lambda_{\infty,i} + \frac{1}{\lambda}\text{Tr}\left(\Sigma_\infty\right) \\
&\geq \sum_{i=1}^{i_\lambda-1} \frac{\lambda_{\infty,i}}{\lambda + \lambda_{\infty,i}} - \frac{1}{\lambda}\sum_{i=1}^{i_\lambda-1} \lambda_{\infty,i} + \frac{1}{\lambda}\text{Tr}\left(\Sigma_\infty\right).
\end{aligned}$$

Therefore, by inequality (55), the convergence of $\lambda_{\infty,i}^{(s)} \to \lambda_{\infty,i}$ for $i \in \{1, \ldots, i_\lambda - 1\}$ as $s \to \infty$, and the second statement in Proposition E, we have

$$\text{plim}_{s\to\infty} \lim_{M\to\infty} \text{Tr}\left(\Sigma_M^{(s)}(\Sigma_M^{(s)} + \lambda I)^{-1}\right) \leq 9\text{Tr}\left(\Sigma_\infty(\Sigma_\infty + \lambda I)^{-1}\right). \tag{56}$$

Combining (45)-(48) with (51), (53), (54), and (56), we establish the counterpart of Theorem B. For given $\epsilon$, $\lambda$, and $\delta$, there exist sufficiently large $s$ and $M$ such that with high probability $1 - \delta$,

$$\begin{aligned}
\mathbb{E}\left[\left\|\overline{g}^{(T)} - g_\rho\right\|_{L_2(\rho_X)}^2\right] &\leq \epsilon + \alpha\lambda^{2r}\|\Sigma_\infty^{-r}g_\rho\|_{L_2(\rho_X)}^2 + \frac{\alpha}{T+1}\left(1 + \frac{1}{\lambda\eta^2(T+1)}\right)\|g_\rho\|_{\mathcal{H}_\infty}^2 \\
&\quad + \frac{\alpha}{T+1}\left(1 + \|g_\rho\|_{L_2(\rho_X)}^2 + \|\Sigma_\infty^{-r}g_\rho\|_{L_2(\rho_X)}^2\right)\text{Tr}\left(\Sigma_\infty(\Sigma_\infty + \lambda I)^{-1}\right),
\end{aligned} \tag{57}$$

where $\alpha > 0$ is a universal constant.

**Proof of Corollary 2.** Since conditions **(A1')** and **(A2')** are special cases of **(A1)** and **(A2)**, we can apply Proposition A to Algorithm 1 for the neural network with the smooth approximation $\sigma^{(s)}$ of ReLU. Hence, by setting $\eta_t = \eta = O(1)$ satisfying $4(6 + \lambda)\eta \leq 1$ and $\lambda = T^{-\beta/(2r\beta+1)}$ where $\beta = 1 + \frac{1}{d-1}$, and by applying $\text{Tr}\left(\Sigma_\infty(\Sigma_\infty + \lambda I)^{-1}\right) = O(\lambda^{-1/\beta})$ (Caponnetto & De Vito, 2007) and

$$\|g_\rho\|_{\mathcal{H}_\infty}, \|g_\rho\|_{L_2(\rho_X)} \leq O\left(\|\Sigma_\infty^{-r}g_\rho\|_{L_2(\rho_X)}\right)$$

because of $\|\Sigma_\infty\|_{\text{op}} \leq O(1)$, we finish the proof of Corollary 2.

# G  APPLICATION TO BINARY CLASSIFICATION PROBLEMS

In this paper, we mainly focused on regression problems, but our idea can be applied to other applications. We briefly discuss its application to binary classification problems. A label space is set to $\mathcal{Y} = \{-1, 1\}$ and a loss function is set to be the squared loss: $\ell(z, y) = 0.5(y - z)^2$. The ultimate goal of the binary classification problem is to obtain the Bayes classifier that minimizes the expected classification error,

$$\mathcal{R}(g) \overset{def}{=} \mathbb{P}_{(X,Y) \sim \rho}[\mathrm{sgn}(g(X)) \neq Y],$$

over all measurable maps. It is known that the Bayes classifier is expressed as $\mathrm{sgn}(g_\rho(X))$, where $g_\rho$ is the Bayes rule of $\mathcal{L}(g) = \mathbb{E}_\rho[l(g(X), Y)]$ (see Zhang (2004); Bartlett et al. (2006)). Therefore, if $g_\rho$ satisfies a margin condition, i.e., $|g_\rho(x)| \geq \exists \tau > 0$ on $\mathrm{supp}(\rho_X)$, then this goal is achieved by obtaining an $\tau/2$-accurate solution of $g_\rho$ in terms of the uniform norm on $\mathrm{supp}(\rho_X)$. That is, the required optimization accuracy on $\|g_{\overline{\Theta}^{(T)}} - g_\rho\|_{L_\infty(\rho_X)}$ to obtain the Bayes classifier depends only on the margin $\tau$ unlike regression problems. Due to this property, averaged stochastic gradient descent in RKHSs can achieve the linear convergence rate demonstrated in Pillaud-Vivien et al. (2018a). To leverage this theory to our problem setting, we consider the following decomposition:

$$\|g_{\overline{\Theta}^{(T)}} - g_\rho\|_{L_\infty(\rho_X)} \leq \|g_{\overline{\Theta}^{(T)}} - \overline{g}^{(T)}\|_{L_\infty(\rho_X)} \tag{58}$$

$$+ \|\overline{g}^{(T)} - g_{M,\lambda}\|_{L_\infty(\rho_X)} \tag{59}$$

$$+ \|g_{M,\lambda} - g_{\infty,\lambda}\|_{L_\infty(\rho_X)} \tag{60}$$

$$+ \|g_{\infty,\lambda} - g_\rho\|_{L_\infty(\rho_X)}. \tag{61}$$

The last term (61) can be made arbitrary small by $\lambda \to 0$ as shown in Pillaud-Vivien et al. (2018a). A term (60) can be bounded in the same manner as the third statement of Proposition E, yielding the convergence to 0 as $M \to \infty$ with high probability. The convergence of (59) was shown in Pillaud-Vivien et al. (2018a) and the convergence of (58) is guaranteed by Proposition A. As a result, we can show the following exponential convergence of the classification error $\mathcal{R}(g)$ for two-layer neural networks with a sufficiently small $\lambda$ as demonstrated in Pillaud-Vivien et al. (2018a).

$$\mathbb{E}[\mathcal{R}(g_{\overline{\Theta}^{(T)}}) - \mathcal{R}(g_\rho)] \leq 2 \exp(-O(\lambda^2 \tau^2 T)).$$

In Nitanda & Suzuki (2019), an exponential convergence was shown for the logistic loss $\ell(z, y) = \log(1 + \exp(-yz))$ as well. Proposition A also holds for the logistic loss with an easier proof than the squared loss because of the boundedness of stochastic gradients of the loss. Hence, their theory is also applicable to the reference ASGD in an RKHS. In summary, (58), (59), and (61) can be bounded by the above argument. However, we note that bounding (60) is not obvious and is left for future work.

