# OpenReview forum: "Optimal Rates for Averaged Stochastic Gradient Descent under Neural Tangent Kernel Regime"
_ICLR.cc/2021/Conference — ICLR 2021 Oral_

### Official Review · AnonReviewer4 · 2020-10-28
**Official Blind Review #4**

**Rating:** 7
**Confidence:** 3

**Review:**

Summary:

The paper focuses on the understanding of neural tangent kernel (NTK), which has been a central topic in deep learning theory recently and plays an important role in characterizing the generalization ability of artificial neural networks. In Particular, the authors derive minimax-optimal learning rates of the averaged stochastic gradient descent method for over-parametrized two-layer neural networks with smooth activation functions. The results are novel and offer insights into the connections between deep learning methods and kernel methods. One difference of this paper from other studies is that the positivity of NTK is not required in the error analysis. Numerical experiments are also illustrated to confirm the theoretical results. The paper is well written and interesting to read. Overall, I vote for accepting.


Concerns:
1. Although the paper considers smooth approximations of the ReLU and also shows the explicit optimal rate, its analysis does not apply to ReLU networks which may be of greater interest to researchers.

2. It looks a little bit strange to me that the regularization is conducted around the initial values. Will this lead to a big difference in the numerical experiments?

3. Is there an explicit form for the lower bound of network width in Theorem 1, i.e., $M_0$?

---

> ### Author Response · Authors · 2020-11-15
> **To Reviewer 4**
>
> Thank you for the review and positive feedback.
>
> **Although the paper considers smooth approximations of the ReLU and also shows the explicit optimal rate, its analysis does not apply to ReLU networks which may be of greater interest to researchers.**
>
> I agree that the analysis of ReLU networks is very important. To do so, for ReLU, we should derive a counterpart of Proposition A. But we focus on smooth activation functions for simplicity in this study.
>
> **It looks a little bit strange to me that the regularization is conducted around the initial values. Will this lead to a big difference in the numerical experiments?**
>
> The regularization on the distance to the initial parameters is rather natural under the NTK regime because it makes the function closer to zero function like the regularization in kernel methods. Moreover, it is also related to the early stopping to some extent, and the practical advantage of this regularization is studied in recent work [W. Hu, Z. Li, and D. Yu].
>
> [W. Hu, Z. Li, and D. Yu] Simple and Effective Regularization Methods for Training on Noisily Labeled Data with Generalization Guarantee.
> https://arxiv.org/abs/1905.11368
>
> **Is there an explicit form for the lower bound of network width in Theorem 1, i.e., $M_0$?**
>
> Yes, $M_0$ is a lower bound on the width, which depends on the number of iterations $T$ (i.e., number of samples) exponentially.  It is somewhat difficult to derive the lower bound in explicit form because of its complexity, but we note that derivation and reducing $M_0$ are not our main focus.
> Please see also comment to Reviewer 3.

---

### Official Review · AnonReviewer2 · 2020-10-28
**Seems very good**

**Rating:** 8
**Confidence:** 2

**Review:**

This paper considers the optimization of a wide two layers neural network (for a regression task) using averaged SGD. The authors consider the Neural Tangent Kernel (NTK) regime. The NTK is a kernel defined using the activation function and the initial distribution of the parameters of the input layer. The RKHS H associated to this NTK is assumed to contain the Bayes predictor. Based on this, the authors derive a convergence rate for the predictor constructed from the T-th iterate of averaged SGD and the Bayes predictor in terms of the L2 distance wrt the distribution of the features. By specifying this bound in terms of the decay of the eigenvalues of the integral operator in H, they obtain an explicit generalization error bound, which is optimal for the class of problems considered in the paper.

It seems that the paper solves an important problem related to the training of neural nets. The paper is rather well written even if some paragraph are hard to understand for a non expert.
For example, several paragraphs of the paper are dedicated to compare the obtained results with those of concurrent work. The level of technicality of these discussions makes the reading experience difficult (e.g. last paragraph of Page 6), often because the discussion happens at a step where the reader is not familiar with the results (e.g. Section 1.2). These paragraphs seem like a discussion with the authors of these concurrent works. I would suggest to gather these discussions at the end of the paper, once the reader understands the results. Moreover, a better approach in my opinion would be to explicitly state the results (mathematically) of these concurrent work. This way, the comparison will be easier.

As a non expert, I believe that the results are new but I cannot be sure because I cannot compare with existing works (see my comment above). I recommend acceptation.


Overall, the paper shows an important result: optimal rate for generalization bounds for 2 layers NN in the NTK regime. The result is well explained but some precision could make the paper even more insightful. For instance, why considering the NTK regime? What is the intuition behind that? How would you define mathematically "the NTK regime" ? I would also like to understand better the relaxation of the positivity of the NTK. Does it have to do with the assumption that the norm of the integral operator is greater than lambda?


Moreover, Proposition A, which is fundamental in the approach, should be stated in the main paper for a better understanding. I think that the authors can move the numerical experiments to the appendix to win some space.




MINOR:

Check the def of excess risk, last equation of Page 3
"negative dependence on" should be "inverse dependence on"
", That is, "
Why is Figure 1 in Section 1? Is it a mistake? It should not be placed here. Moreover it is not commented in the text.

---

> ### Author Response · Authors · 2020-11-15
> **To Reviewer 2**
>
> Thank you for reviewing our paper and for helpful suggestions.
>
> **As a non expert, I believe that the results are new but I cannot be sure because I cannot compare with existing works (see my comment above). I recommend acceptation. Moreover, Proposition A, which is fundamental in the approach, should be stated in the main paper for a better understanding.**
>
> Although it is somewhat difficult to summarize existing results mathematically because they consider slightly different assumptions and problem settings each other, there is a common thought that neural networks under a certain setting (i.e., NTK regime) can behave similarly to the kernel method with NTKs.
> The idea behind this is the positive NTK leads to a rapid decay of the training loss and an NTK is almost fixed by overparameterization, hence learning dynamics can be localized around the initial point, resulting in the equivalence between neural networks and kernel methods through a linear approximation of neural networks. This story is simply stated in the introduction.
>
> However, in previous studies, such connections were provided regarding on training dataset or sample-wise high probability. Hence, we cannot fully utilize the beneficial property of kernel methods, and we overcome this issue in this study by providing a new connection between neural networks and NTKs.  The key is Proposition A and we have added its informal version to Section 1.2 to highlight the difference from the previous result. (Please see also comment to Reviewer 1).
>
> **For instance, why considering the NTK regime? What is the intuition behind that? How would you define mathematically "the NTK regime"? I would also like to understand better the relaxation of the positivity of the NTK. Does it have to do with the assumption that the norm of the integral operator is greater than $\lambda$?**
>
> By convention, the NTK regime means the setting of neural networks where the above connection can be established with overparameterization. A typical setup is two-layer neural networks of width $M$ with the output scale $1/\sqrt{M}$.
>
> The idea behind the relaxation of the positivity of the NTK is that we relax the requirement of localization. Indeed, Proposition A says that the overparameterization increase time stayed in the NTK regime. And we acquire a more useful connection to show a faster convergence rate.
>
> The assumption on $\lambda$ is due to a technical reason for bounding the variance caused by the sampling of initial parameters. (See Proposition 10 in [A. Rudi and L. Rosasco (2017)]). In addition, this assumption is essentially considered as a lower bound on the number of iteration (i.e., number of samples) because $\lambda$ is chosen as $\lambda \sim T^{-p}$ (where $p$ is an appropriate value). Such a restriction on the number of samples is rather common in the literature of random feature approximation (e.g., [A. Rudi, L. Carratino, and L. Rosasco (2017)] and [A. Rudi and L. Rosasco (2017)]). We note that the norm of the integral operator depends only on the data distribution and activation function.
>
> [A. Rudi and L. Rosasco (2017)] Generalization Properties of Learning with Random Features.
> https://arxiv.org/abs/1602.04474
>
> [A. Rudi, L. Carratino, and L. Rosasco (2017)] FALKON: An Optimal Large Scale Kernel Method.
> https://arxiv.org/abs/1705.10958
>
> **Check the def of excess risk, last equation of Page 3.**
>
> We have fixed this equation.  (The noise term $\sigma^2 = L(g_\rho)$ is moved to the left hand side).
>
> **Why is Figure 1 in Section 1? Is it a mistake? It should not be placed here.**
>
> Figure 1 is empirical evidence for the degeneration of NTKs to motivate convergence analysis not relying on the positivity of the NTK.  This figure was cited in the third paragraph of the introduction.

---

> > ### Comment · AnonReviewer2 · 2020-11-23
> > **Thanks**
> >
> > Thanks for your answers and your explanations, I keep my score as it is.

---

### Official Review · AnonReviewer1 · 2020-11-03
**Optimal Rates for Averaged Stochastic Gradient Descent under Neural Tangent Kernel Regime**

**Rating:** 8
**Confidence:** 4

**Review:**

Here is the review of the article: ``Optimal Rates for Averaged Stochastic Gradient Descent under Neural Tangent Kernel Regime''.

**SUMMARY**

The authors show that under the Neural Tangent Kernel Regime investigated lately, averaged SGD achieves optimal rates in the attainable case.  Note that this result is not a *plug and play* based on the current kernelized-SGD litterature: the difficulty the authors achieve to overcome is the fact that they could bound the difference between the dynamics of SGD on the neural networks and on the neural tangent kernel. This is stated in Proposition A and it represents the novelty of the article

Moreover, they give an explicit representation of the capacity condition (decrease rate of the eigenvalues of the covariance matrix) in the cases where they have a smooth approximation of the ReLu.

**Clarity**

The paper is outstandly clear. Indeed, despite the fact that optimality in RKHS can be technical to introduce, I found the paper very clearly presented and well motivated. It was very pleasant and smooth to read. The references are also both precise and sufficient to understand well the problem.

The only default of the paper is that, in my opinion, the authors do not stress enough their contribution and their novelty. Indeed, despite Proposition A and the sketch of the proof, the article consists in a *plug and play* result on averaged SGD. The authors should then stress the outline of the proof (going to a M-approximation of the Neural Tangent RKHS) and comparing the dynamics on these.

**Quality and Originality**

The paper is not super original, and as accustomed to this literature, there is no surprise seeing this result. However, the quality of the paper is undeniable and fills the gap between optimality of kernel methods and the NTK literature. I thank the authors to have done it very clearly.



**Comments**

-My main comment is about the fact that except from Proposition A, this article is a *plug and play* one. This proposition, and the full sketch of the proof should be emphasized as they consist on the true novelty of the work.

-Three remarks concerning the plots :

1-*Minor comment.* They should be bigger.
2-*Minor comment.* Figure 1 should illustrate the fact that $\beta = 1 + \frac{1}{d-1}$. I suggest a log-log plot.
3-*Intermediate comment.* I do not really see how exactly the discussion of the experimental part illustrates really the result. The discussion is fairly interesting, but I really would like to see a theoretical proposition showing that taking the two layers is better for learning.


*** Conclusion***

Yet the fact remains that, I would really like this article to be published when the commentaries of the reviewers will be taken into accounts.

---

> ### Author Response · Authors · 2020-11-15
> **To Reviewer 1**
>
> Thank you for your careful reading and thoughtful comment.
>
> **My main comment is about the fact that except from Proposition A, this article is a plug and play one. This proposition, and the full sketch of the proof should be emphasized as they consist on the true novelty of the work.**
>
> As you said, the key in our theory is Proposition A which provides a new connection between a neural network and NTK. Remarkable points of Proposition A are providing a connection with respect to the uniform norm and not relying on the positivity of NTK, unlike existing studies. Instead of the positivity, we show that the overparameterization increases time stayed in the NTK regime. To emphasize our contribution, we have added this argument with a simplified version of Proposition A to Section 1.2.
>
> **I do not really see how exactly the discussion of the experimental part illustrates really the result. The discussion is fairly interesting, but I really would like to see a theoretical proposition showing that taking the two layers is better for learning.**
>
> We would like to explain the idea behind the experiment. We note that eigenvalues of NTK of the two-layer network are roughly expressed as follows (see equation (44) in which $R$ is set to $1$):
>
> $\lambda_{k}^{(1)}+ R^2(\gamma^2\lambda_{k}^{(2)} + \lambda_{k+1}^{(2)} + \lambda_{k-1}^{(2)}).$
>
> Here, $\lambda_k^{(1)}$ and $\lambda_k^{(2)}$ are eigenvalues of NTKs of output layer and input layer, respectively. Interestingly, both layers share the same eigenfunctions and they are complementary: $\lambda_k^{(1)}=0$ when $k$ is odd and $\lambda_k^{(2)}=0$ when $k$ is even except for $k=1,2$ under the setting of Section 3.3. This means these two NTKs are almost orthogonal to each other when taking $R \sim 1/\gamma$ and $\gamma \rightarrow \infty$. This is the reason why we set $R=1/(20\sqrt{2})$ and $\gamma=10\sqrt{2}$ and misspecification happened in single-layer learning. In addition, we can find an NTK of each single-layer is included in the NTK of two-layers.

---

### Official Review · AnonReviewer5 · 2020-11-04
**A nice result**

**Rating:** 8
**Confidence:** 5

**Review:**

This paper analyzed the averaged SGD for overparameterized two-layer NNs for regression problems. Particularly, they show that the averaged SGD can achieve the minimax optimal convergence rate, with the global convergence guarantee. To achieve, they propose a new parameter which captures the ``complexities’’ of the target function and the RKHS associated with the NTK.

The paper is well-written, and the result looks very interesting. I am tending to accept the paper. This paper is a theory, so experiments are a plus. If the authors can address some of my comments, I am tending to increase the score of the paper.



Here are some comments about writing.

I believe Assumption 1 and 2 are reasonable. But each statement is just math, it is good to write a 2~4 words to summarize each A1, A2 .. Also several math statements highly replied on the definitions, and it is hard to find them in the paper.

After Assumption 1, in the next page, page 5, there is a Remark that has 4 bullets, maybe write Ai at the beginning of each bullet.

Algorithm 1 requires more words and explanation, it is hard to understand this algorithm in the following sense : the size of each matrix/vector is not mentioned and hard to find them in the paper.

In page 6, is it possible to simplify the statement of Theorem 1 a bit? e.g. write a simplified version here, and put the full version in appendix.

In Theorem 1, what is M_0? Is that over-parameterization size? Is that polynomial in parameters or exponentially in parameters? (This is not major point of the paper, I am just curious about the bound)

In page 4, Eq. (2), is it possible to consider a simple model where gamma = 0? This is quite common in previous work.

In appendix, e.g. page 30, the last step of many equations use ->0. I don’t follow the meaning of this notation. Is that possible to avoid it?

This paper is focusing on average SGD, is there any intuition why non-average SGD won’t give the similar result?

I felt the following paper is highly close to this work, and should be cited and discussed more deeply. Usually optimization has two parts, one is the number of iterations, and the other is cost per iteration. This paper focused on improving the number of iterations. The following paper improved the cost per iteration, in the NTK overparameterized regime.
Training (Overparametrized) Neural Networks in Near-Linear Time
Jan van den Brand, Binghui Peng, Zhao Song, Omri Weinstein


Minor comments

In page 1, second paragraph, the place cited Du et al. 2019b, Allen-Zhu et al. 2019 and Du et al. 2019a.

The following two papers should also be cited

Zeyuan Allen-Zhu, Yuanzhi Li, and Zhao Song. On the convergence rate of training recurrent neural networks.  [This paper shows the result for recurrent neural networks. Note that RNN is  a harder case, In deep neural networks the weight matrices in different layers are different. However in RNN, the weights matrix are the same over all the layers]

Zhao Song and Xin Yang. Quadratic suffices for over-parametrization via matrix chernoff bound.
[This paper improved the over-parameterization bound from m >= n^6 (Du et al. 2019b) to m >= n^4, where m is the width of a neural network, and n is the number of input data points.]

In page 2, the first paragraph, the place cited Du et al. 2019b, Arora et al. 2019a, Weinan et al. 2019, Arora et al. 2019b, Lee et al,. 2019.

The following papers should also be cited

Jason D Lee, Ruoqi Shen, Zhao Song, Mengdi Wang, and Zheng Yu. Generalized leverage score sampling for neural networks. [Arora et al. paper shows a connection between neural networks with neural tangent kernel regression. This paper generalizes the Arora et al result, and shows the connection between regularized neural networks with neural tangent kernel ridge regression.]

Similarly, in page 5, some citations should be added.

Small typos:
The last paragraph, Page 2 “the key to show” -> “the key to showing”
The third paragraph, Page 3 “which enable” -> “which enables”
The fifth paragraph, Page 4 “ A stochastic gradient descent” -> “Stochastic gradient descent”
The third paragraph Page 5 “a neural networks” -> “a neural network”
The third paragraph Page 6 “arbitrary small” -> “arbitrarily small”
The first paragraph Page 8 “the single-layer learning” -> “single-layer learning”

---

> ### Author Response · Authors · 2020-11-15
> **To Reviewer 5**
>
> Thank you for the thorough reading of our paper and valuable suggestions. We have revised our paper by taking into account your suggestions (about the description of algorithm and assumptions with subsequent remark), which have further improved the quality of the paper.
>
> **In Theorem 1, what is $M_0$? Is that over-parameterization size? Is that polynomial in parameters or exponentially in parameters? (This is not major point of the paper, I am just curious about the bound)**
>
> $M_0$ is a bound on overparameterization, which can exponentially increase depending on the number of examples (i.e., iterations of ASGD). Please see also comment to Reviewer 3.
>
> **In page 4, Eq. (2), is it possible to consider a simple model where $\gamma = 0$? This is quite common in previous work.**
>
> Thank you for raising this point. The reason why we assume $\gamma > 0$ is to keep the minimax optimality of the convergence rate under a specific setting in Section 3.3 where the target function is specified by ReLU-NTK. The condition $\gamma > 0$ is closely related to Assumption (A4): $\lambda_i = \Theta(i^{-\beta})$ for this setting. We first note that the convergence rate of $O(T^{-2r\beta/(2r\beta+1)})$ in Theorem 1 and Corollary 1, 2 always holds even though (A4) is relaxed to $\lambda_i = O(i^{-\beta} )$ and $\gamma = 0$ as long as the other conditions hold. However, to make the rate of $O(T^{-2r\beta/(2r\beta+1)})$ minimax optimal, the lower bound on $\lambda_i$: $\lambda_i=\Omega(i^{-\beta})$ is also needed.  Under the setting in Section 3.3, this lower bound does not hold if we set $\gamma=0$, hence the obtained rate is no longer minimax optimal.
>
> We think the above argument is quite informative to the readers, so we have made the following small changes in the paper:
> - In (A2): $\gamma \in (0,1]$ -> $\gamma \in [0,1]$.
> - We have added a comment to the remark on (A4): ``Theorem 1 and Corollary 1, 2 hold even though the condition in (A4)  is relaxed to $\lambda_i = O(i^{-\beta})$ and the lower bound $\lambda_i=\Omega(i^{-\beta})$ is necessary only for making obtained rates minimax optimal.”
>
> **In appendix, e.g. page 30, the last step of many equations use $\rightarrow 0$. I don’t follow the meaning of this notation. Is that possible to avoid it?**
>
> Since we have to show the convergence of several quantities (kernel, covariance operator, and minimizers) with respect to $s \rightarrow \infty$ and $M\rightarrow \infty$, these equations are probably inevitable. Note that notations plim and $\xrightarrow{p}$ mean the convergence in probability and we used the uniform law of large number and Bernstein’s inequality to random operators for showing this convergence. We have made the proof of this proposition E a little more detailed.
>
> **This paper is focusing on average SGD, is there any intuition why non-average SGD won’t give the similar result?**
>
> The key to improving the performance of the vanilla SGD is to combine variance reduction methods with them because SGD usually suffers from the variance of stochastic gradient. Hence there are many studies analyzing variance reduction methods and showing their superiority. An averaging method is also one of such methods, which works very effectively especially for strongly convex problems. Indeed, most studies that show the optimality of optimization methods for nonparametric regression focus on variants of average SGD (e.g., [Dieuleveut and Bach (2016)], [Pillaud-VIvien et al. (2018a,2018b)], [Mücke et al. (2019)], etc.).
>
> **Additional related papers.**
>
> Thank you for bringing up several related papers. We have cited them according to your suggestion.  Concretely, [Brand et al. (2020)] together with [Zhang et al. (2019)] and [Cai et al. (2019)] which show the convergence of the second-order methods are cited in the first paragraph of Section 1.3. [Zeyuan Allen-Zhu et al. (2019b)] which shows the global convergence for RNN is cited in the second paragraph of Section 1, and [Song and Yang (2019)] is also cited in the second paragraph of Section 1.3.  [Lee et al. (2020)] which generalizes the connection between a neural network and NTK is cited in Section 1.2 and the other appropriate places.
>
> **Small typos.**
>
> Thank you for pointing out these typos and we have fixed them.

---

### Official Review · AnonReviewer3 · 2020-11-05
**Technically good paper**

**Rating:** 7
**Confidence:** 2

**Review:**

Summary:

This paper considers the convergence property of averaged stochastic gradient descent on a overparameterized two-layer neural networks for a regression problem. This paper is the first to achieve the optimal convergence rate under the NTK regime. They show that smooth target functions efficiently specified by the NTK are learned rapidly at faster convergence rate.


##########################################################################Pros:

+ The paper is technically sound. It adapt the neural networks and the RKHSs theory into the NTK regime and the proof techniques are different from existing literatures.

+ It achieves the minimax optimal convergence rate of $O(T^{-2r\beta/2r\beta + 1})$ which is always faster than $O(T^{-1/2})$.

+ This work shows the connection between RKHS and NN in term of the $L_{\infty}(\rho_X)$ norm while the convergence result does not need the positivity of the NTK


##########################################################################
Cons:

- The writing of this paper is not clear enough, for example $L_2(\rho(x))$ appeared without any explaination
- The practical choice of M in their experiment is 2e4, which is impractical in NN tasks


##########################################################################
Overall, I vote for accepting. This paper combine the convergence analysis of averaged stochastic gradient descent on kernel methods with the connection of kernel method with neural network to derive an optimal convergence rate of NN in NTK regime while the proof technique is novel. My major concern is about the practical influence of this paper to the theory of deep learning training, as the width of the output layer is required to be large.

#########################################################################

Questions:

In Assumption A3, why the parameter $r$ has to be in the range [1/2, 1]
Is the averaging mechanism at the output of the algorithm beneficial to the rate in Theorem 1?

---

> ### Author Response · Authors · 2020-11-15
> **To Reviewer 3**
>
> Thank you for the positive feedback and comments.
>
> **The writing of this paper is not clear enough, for example, $L_2(ρ_X)$ appeared without any explanation.**
>
> We omitted the definition of $L_2$-space. The meaning of $\rho_X$ was commented in Section 1.2 and the first paragraph of Section 2. We have added more comments and definitions where these notations first appeared.
>
> **The practical choice of M in their experiment is 2e4, which is impractical in NN tasks. My major concern is about the practical influence of this paper to the theory of deep learning training, as the width of the output layer is required to be large.**
>
> We would like to emphasize that an arbitrarily large network width $M$ can be required to achieve an arbitrarily small excess risk in terms of population risk because the target function (Bayes rule) exists in an infinite-dimensional RKHS associated with the NTK. In other words, the divergence of $M$ as $T\rightarrow \infty$ (which means the convergence of the excess risk) is inevitable for the regression problem under this setting. That is why we used a large network in experiments to verify our theory.
> Although there is a possibility to improve the dependency of $M$ on the number of samples which corresponds to $T$, it is not the main focus of this study. In addition, we note that for the binary classification problem under the low noise settings (see the last section in Appendix), the size of $M$ can be significantly reduced to achieve the desired classification error because the classification error converges at an exponential rate by averaged SGD (ASGD), which means the required order of $T$ is logarithmic with respect to a required error.
>
> **In Assumption A3, why the parameter r has to be in the range [1/2, 1].**
>
> On one hand the condition $r < 1/2$ means that the target function (Bayes rule) lies outside an RKHS associated with NTK (i.e., misspecification of the target), on the other hand, the condition $r>1$ means that the target function is very smooth. Interestingly, learning in both settings is known to be difficult and ASGD is suboptimal for the kernel method. Recently, there are several attempts ([Pillaud-Vivien et al. (2018b)] and [Mücke et al. (2019)]) to overcome this limitation under a certain setting but they require modifications of ASGD, hence we focus on the condition $r\in [1/2,1]$ in this study.
>
> **Is the averaging mechanism at the output of the algorithm beneficial to the rate in Theorem 1?**
>
> In general, for learning problems in an RKHS, the convergence rate of ASGD is known to be faster than that for SGD. Moreover, the practical performance of ASGD is certainly better because of the variance reduction of the noise of stochastic gradients while SGD usually suffers from them.

---

### Author Response · Authors · 2020-11-15
**To all reviewers.**

We thank all reviewers for the positive feedback and helpful comments. We have revised our paper by taking into account some of their suggestions. (During the rebuttal phase the page limit is 9 pages). The major changes are summarized below.
1. We have elaborated on some description of the mathematical definition, assumptions, and algorithm.
2. We have made a small change: In Assumption (A2), $\gamma \in (0,1]$ → $\gamma \in [0,1]$, and added a remark. See also comment to Reviewer 5.
3. We have added some citations.
4. We have added a simplified version of Proposition A to the introduction in order to highlight the contribution. See also comment to Reviewer 1.

---

### Decision · Program_Chairs · 2021-01-07
**Final Decision**

**Decision:**

Accept (Oral)

**Comment:**

The paper presents some exciting results on the convergence of averaged SGD for overparameterized two-layer neural networks. The AC and reviewers all agree that the contributions are significant and well presented, and appreciate the author feedback to the reviews. The corresponding revisions on assumptions and references, and the added simplified proposition in the introduction have nicely improved the manuscript.